# Bambusicolous Fungi in Pleosporales: Introducing Four Novel Taxa and a New Habitat Record for *Anastomitrabeculia didymospora*

**DOI:** 10.3390/jof8060630

**Published:** 2022-06-13

**Authors:** Rungtiwa Phookamsak, Hongbo Jiang, Nakarin Suwannarach, Saisamorn Lumyong, Jianchu Xu, Sheng Xu, Chun-Fang Liao, Putarak Chomnunti

**Affiliations:** 1School of Science, Mae Fah Luang University, Chiang Rai 57100, Thailand; rphookamsak@outlook.com (R.P.); hongbo-j@hotmail.com (H.J.); fang04271@hotmail.com (C.-F.L.); 2Department of Economic Plants and Biotechnology, Yunnan Key Laboratory for Wild Plant Resources, Kunming Institute of Botany, Chinese Academy of Sciences, Kunming 650201, China; jxu@mail.kib.ac.cn; 3Honghe Center for Mountain Futures, Kunming Institute of Botany, Chinese Academy of Sciences, Honghe 654400, China; hambugerking123@gmail.com; 4East and Central Asia Regional Office, World Agroforestry Centre (ICRAF), Kunming 650201, China; 5Centre for Mountain Futures (CMF), Kunming Institute of Botany, Kunming 650201, China; 6Center of Excellence in Fungal Research, Mae Fah Luang University, Chiang Rai 57100, Thailand; 7Research Center of Microbial Diversity and Sustainable Utilization, Faculty of Sciences, Chiang Mai University, Chiang Mai 50200, Thailand; suwan_461@hotmail.com (N.S.); scboi009@gmail.com (S.L.); 8Department of Biology, Faculty of Science, Chiang Mai University, Chiang Mai 50200, Thailand; 9Academy of Science, The Royal Society of Thailand, Bangkok 10300, Thailand; 10Innovative Institute of Plant Health, Zhongkai University of Agriculture and Engineering, Haizhu, Guangzhou 510225, China

**Keywords:** Dothideomycetes, morphology, multigene phylogeny, Parabambusicolaceae, Pyrenochaetopsidaceae, taxonomy, Tetraploasphaeriaceae

## Abstract

While conducting a survey of bambusicolous fungi in northern Thailand and southwestern China, several saprobic fungi were collected from dead branches, culms and twigs of bamboos, which were preliminarily identified as species belonging to Pleosporales (Dothideomycetes) based on a morphological approach. Multigene phylogenetic analyses based on ITS, LSU, SSU, *rpb2*, *tef1-α* and *tub2* demonstrated four novel taxa belonging to the families Parabambusicolaceae, Pyrenochaetopsidaceae and Tetraploasphaeriaceae. Hence, *Paramultiseptospora bambusae* sp. et gen. nov., *Pyrenochaetopsis yunnanensis* sp. nov. and *Tetraploa* *bambusae* sp. nov. are introduced. In addition, *Anastomitrabeculia didymospora* found on bamboo twigs in terrestrial habitats is reported for the first time. Detailed morphological descriptions and updated phylogenetic trees of each family are provided herein.

## 1. Introduction

Bamboo is one of the most useful perennial woody grasses that contains the highest amount of living biomass [1]. It belongs to the subfamily Bambusoideae, family Poaceae, comprising about 127 genera, with approximately 1680 species, covering around 25 million hectares in tropical, subtropical, and mild temperate regions of Africa, America, Asia and Oceania, but it is rarely found in Europe [1,2,3,4,5]. Bamboo is important for ecological and socioeconomic sustainability throughout the world. Bamboo forests are crucial for the environmental benefits and climate change mitigation; they are sustainable in soil erosion control, carbon sequestration, soil and water conservation, windbreaks and shelterbelts, land rehabilitation, as well as releasing negative oxygen ions [4,6,7,8,9]. Bamboo has also been utilized for traditional Chinese medicine, food sources, furniture and building construction, musical instruments, paper and textile industries, reinforcing fibers, as well as feedstock for bioethanol and biomethane productions [1,4,7,9,10]. Bamboo is considered to be an eco-friendly plant, but there are also potential problems associated with bamboo’s cultivations such as decreasing biodiversity and soil fertility, soil and water loss, and surface water pollution [7].

Bamboo are highly diverse, distributed worldwide, contain high biomass and are a sustainable carbon source; hence, they host a high diversity of fungi [3,8,11,12,13]. Study into bambusicolous fungi has been carried out since the 18^th^ century, which was first started by Léveillé [14]. Subsequently, many mycologists have described an increasing number of fungi on bamboo, especially ascomycetes [3,11,12,13,15,16,17,18,19]. Recently, more than 1300 fungi have been reported on bamboo, including 150 basidiomycetes and 800 ascomycetes, of which 350 species are reported as asexual morphs [13]. However, many bambusicolous fungi remain poorly clarified in taxonomic classification due to the lack of molecular–phylogenetic approaches [3,12].

Over the last two decades, taxonomic studies of bambusicolous fungi have become an interesting research topic for many Asian mycologists [3,12,13,20,21,22,23,24,25,26,27,28,29,30,31,32,33,34,35,36,37,38,39,40,41,42,43,44,45,46,47,48,49,50,51,52,53,54]. The taxonomic classification of bambusicolous fungi based on molecular phylogeny was initially carried out by Tanaka et al. [12], who introduced a novel family Tetraplosphaeriaceae to accommodate tetraploa-like taxa in Pleosporales. Noteworthily, major studies of bambusicolous ascomycetes with modern taxonomic treatments have been carried out by Dai et al. [3], Liu et al. [31,32], Phookamsak et al. [39], and Tanaka et al. [27]. Up to now, more than 175 bambusicolous ascomycetes have been described based on morphological and phylogenetic evidence [13,42,43,44,45,46,47,48,49,50,51,52,53,55,56,57,58,59,60,61,62,63,64,65,66,67,68,69,70]. However, there have been few thorough studies into the phylogeny-based taxonomy of bambusicolous fungi preceding 2002 [11], causing more than 80% of bambusicolous fungi to lack molecular data that could clarify their phylogenetic placement.

Many families in Pleosporales were initially introduced to accommodate bambusicolous fungi mainly such as Anastomitrabeculiaceae [71], Astrosphaeriellaceae [39], Bambusicolaceae [72], Occultibambusaceae [3], Parabambusicolaceae [27], Pseudoastrosphaeriellaceae [39], Roussoellaceae [32], Shiraiaceae [73] and Tetraploasphaeriaceae [12]. Furthermore, bambusicolous fungi are also distributed among many families in Pleosporales such as Aigialaceae, Dictyosporiaceae, Didymosphaeriaceae, Halotthiaceae, Lentitheciaceae, Ligninsphaeriaceae, Longipedicellataceae, Lophiotremataceae, Periconiaceae, Phaeosphaeriaceae, and Pleosporaceae [27,31,41,49,52,74,75,76,77,78,79,80,81], suggesting that bambusicolous fungi are diverse in Pleosporales. In this study, we also found the bambusicolous fungi in Pleosporales with the intention of providing a better understanding of their taxonomy placement. The aim of this study is to introduce four novel taxa of bambusicolous fungi in Pleosporales based on the morpho-molecular approach.

## 2. Materials and Methods

### 2.1. Collection, Examination, Isolation, and Preservation

Samples were collected from the dead branches, culms, and twigs of bamboo in Chiang Mai and Chiang Rai Provinces of Thailand in 2011 and Yunnan Province of China in 2021. The samples were stored in paper bags and brought to the laboratory for observation and examination. Fungal fruiting bodies on host substrates were observed under an Olympus SZ61 series stereo microscope, and a centrum was mounted in sterilized distilled water on a clean slide for examination and captured under a Nikon ECLIPSE Ni compound microscope connected to a Nikon DS-Ri2 camera. Cotton blue was added to observe the fungal centrum, and Indian ink was used to check the mucilaginous sheath covering the ascospores. Morphological features were measured using Tarosoft (R) Image FrameWork version 0.9.7. Photographic plates were edited and combined in Adobe Photoshop CS6 software (Adobe Systems Inc., San Jose, CA, USA). The permanent slides were prepared by adding lacto-glycerol and sealed by nail polish and deposited with herbarium specimens at the Herbarium of Cryptogams Kunming Institute of Botany Academia Sinica (KUN-HKAS), China and the herbarium of Mae Fah Luang University, Chiang Rai, Thailand (MFLU).

Pure cultures were obtained from single-spore isolation based on a spore suspension technique [82]. Germinated ascospores were aseptically transferred to potato dextrose agar (PDA) and cultivated under normal light at 20–25 °C. Fungal colonies were observed and recorded after one week and four weeks. The asexual morph that sporulated in vitro was observed and examined after two months. Axenic living cultures were deposited in the Mae Fah Luang University Culture Collection (MFLUCC) and the Culture Collection of Kunming Institute of Botany (KUMCC). The newly described taxa were registered in Index Fungorum (http://www.indexfungorum.org/names/IndexFungorumRegister.htm; accessed on 13 May 2022).

### 2.2. DNA Extraction, Amplification, and Sequencing

Fresh mycelia were scraped from fungal colonies growing on PDA for a month and stored in a 1.5 mL sterilized microcentrifuge tube in an aseptic condition. Fungal genomic DNA was extracted by using Biospin Fungus Genomic DNA Extraction Kit (BioFlux^®^, Hangzhou, China) following the manufacturer’s instructions (Hangzhou, China). Fungal genomic DNA was also extracted from fruiting bodies directly in case the fungi could not germinate on PDA using a Forensic DNA Kit (Omega^®^, Norcross, GA, USA). The generated fungal genomic DNA was stored at 4 °C for PCR amplification and duplicated at −20 °C for long-term storage.

Fungal genomic DNA was amplified by polymerase chain reaction (PCR) using informative phylogenetic markers of each family, including the internal transcribed spacers (ITS1-5.8S-ITS2), the 28S large subunit rDNA (LSU), the 18S small subunit rDNA (SSU), the partial RNA polymerase second largest subunit (*rpb2*), the translation elongation factor 1-alpha (*tef1-α*) and β-tubulin (*tub2*). The forward and reverse primer pairs ITS5 and ITS4 [83], LR0R and LR5 [84], NS1 and NS4 [83], fRPB2-5F and fRPB2-7cR [85], EF1-983F and EF1-2218R [86], and T1 and BT2B [87,88] were used to amplify the PCR fragments of these genes, respectively. Components of the PCR reaction mixture and the PCR thermal cycle program for ITS, LSU, SSU, *rpb2*, and *tef1-α* genes followed the condition described in Jiang et al. [50]. The PCR thermal cycle program for *tub2* was set up initially at 94 °C for 3 min, followed by 35 cycles of denaturation at 94 °C for 30 s, annealing at 52 °C for 40 s, elongation at 72 °C for 1 min, a final extension at 72 °C for 10 min, before being held at 4 °C. PCR products were sent to TsingKe Biological Technology (Beijing) Co., Ltd., Beijing, China for purification and sequencing. The quality of the Sanger DNA sequences and sequence consensus from forward and reverse directions were checked and compiled manually in BioEdit v. 7.2.3 [89].

### 2.3. Sequence Alignment and Phylogenetic Analyses

The generated ITS sequences of the new isolates were used to search the related fungal group via the nucleotide BLAST search tool in the NCBI website (https://blast.ncbi.nlm.nih.gov/Blast.cgi; accessed on 5 December 2021). The nucleotide BLAST searches of the ITS sequence showed that the newly generated sequences had the closest similarity with species in families Anastomitrabeculiaceae, Parabambusicolaceae, Pyrenochaetopsidaceae and Tetraploasphaeriaceae. Thus, sequences generated from this study were analyzed with representative taxa in Anastomitrabeculiaceae, Parabambusicolaceae, Pyrenochaetopsidaceae and Tetraploasphaeriaceae, which were retrieved from GenBank based on recent publications (Table 1). Individual gene alignments were performed and improved manually where necessary using MEGA 7 [90]. Ambiguous sites were excluded from the alignment. Improved individual gene alignments were prior analyzed by maximum likelihood (ML) analysis using RaxmlGUI version 7.3.0 [91]. After checking the tree topologies of every individual gene alignment for congruence, the combined gene dataset of each family was analyzed based on Bayesian inference (BI), maximum likelihood (ML) and maximum parsimony (MP) analyses.

The evolutionary model of nucleotide substitution analysis was selected independently for each locus using MrModeltest 2.3 [92]. The best-fit model under the Akaike Information Criterion (AIC) of each locus was shown in Table 2. Bayesian inference (BI) was analyzed using MrBayes on XSEDE v. 3.2.7a via the CIPRES Science Gateway v. 3.3 [93]. Posterior probabilities (PP) [94,95] were determined by Markov Chain Monte Carlo sampling (MCMC). Two parallel runs with six simultaneous Markov chains were run for 1–2 million generations and stopped automatically when the average standard deviation of split frequencies reached below 0.01. Trees were sampled every 100th generation. The MCMC heated chain was set with a “temperature” value of 0.15. All sampled topologies beneath the asymptote (25%) were discarded as part of the burn-in procedure, and the remaining trees were used for calculating posterior probabilities in the majority rule consensus tree. Maximum likelihood (ML) was analyzed in RaxmlGUI version 7.3.0 [91] using the default algorithm of the program from a random starting tree for each run that was adjusted by setting up the GTR + GAMMAI model of nucleotide substitution with 1000 rapid bootstrap replicates. Maximum parsimony was analyzed by PAUP v. 4.0b10 [96] using the heuristic search function with 1000 random stepwise addition replicates and tree bisection-reconnection (TBR) as the branch-swapping algorithm. Maxtrees were set up to 1000, and a zero of maximum branch length was collapsed. All characters were unordered and of equal weight, and gaps were treated as missing data. Significant parsimonious trees were determined by Kishino–Hasegawa tests (KHT) [97]. All equally parsimonious trees were saved. Clade stability was estimated by bootstrap (BS) support values with 1000 replicates, each with 10 replicates of random stepwise addition of taxa [98]. Descriptive tree statistics viz. tree length (TL), consistency index (CI), retention index (RI), relative consistency index (RC) and homoplasy index (HI) were calculated.

Phylograms were visualized on FigTree v. 1.4.0 [99], and layouts of trees were drawn in Microsoft Office PowerPoint 2016 (Microsoft Inc., Redmond, WA, USA). The newly generated sequences in this study are deposited in GenBank (Table 1). The final alignment and tree were submitted in TreeBASE (https://www.treebase.org; accessed on 25 March 2022) under submission ID: 29589 (A1: Anastomitrabeculiaceae), 29590 (A2: Parabambusicolaceae), 29592 (A3: Pyrenochaetopsidaceae), and 29593 (A4: Tetraploasphaeriaceae).

## 3. Results

### 3.1. Phylogeny

Four phylogenetic analyses were conducted to resolve phylogenetic relationships of taxa in Anastomitrabeculiaceae (Analysis 1), Parabambusicolaceae (Analysis 2), Pyrenochaetopsidaceae (Analysis 3), and Tetraploasphaeriaceae (Analysis 4) as follows:

Analysis 1: Taxa in Anastomitrabeculiaceae were analyzed with related taxa in families Halojulellaceae and Neohendersoniaceae based on a combined LSU, SSU, ITS, and tef1-α DNA sequence dataset. The Data matrix comprised 20 taxa, of which two species in Massarinaceae (*Helminthosporium aquaticum* MFLUCC 15-0357, and *Massarina eburnea* CBS 473.643) were selected as the outgroup taxa. The dataset consists of 3471 total characters, including gaps (LSU: 1–870 bp, SSU: 871–1900 bp, ITS: 1901–2499 bp, *tef1-α*: 2500–3471 bp). The best scoring RAxML tree is presented in Figure 1 with a final ML optimization likelihood value of −10679.529226 (ln). RAxML analysis yielded 750 distinct alignment patterns, and the proportion of gaps and completely undetermined characters in this alignment was 21.29%. The proportion of invariable sites I = 0.402038 and the gamma distribution shape parameter alpha = 0.46716. The Tree-Length = 0.871849 with estimated base frequencies were as follows: A = 0.242670, C = 0.243741, G = 0.268897, T = 0.244693, and substitution rates AC = 1.181725, AG = 2.821009, AT = 1.536446, CG = 0.754762, CT = 7.660341, GT = 1.000000. The maximum parsimonious dataset consisted of 3471 characters, with 2722 characters being constant (proportion = 0.784212), 212 variable characters being parsimony-uninformative and 537 characters being parsimony-informative. The parsimonious analysis yielded six parsimonious trees, of which the first parsimonious tree was selected as the best tree for the Kishino–Hasegawa test (TL = 1178, CI = 0.778, RI = 0.845, RC = 0.657, HI = 0.222). Bayesian analysis yielded 10,001 trees from one million runs, of which 7501 were sampled. Bayesian posterior probabilities (BYPP) from MCMC were evaluated with the final average standard deviation of split frequencies = 0.004119. 

The phylogenetic results based on maximum likelihood (ML), maximum parsimony (MP) and Bayesian inference analyses (Figure 1) showed overall similar tree topologies. Two new strains of *Anastomitrabeculia didymospora* (MFLUCC 11-0197, MFLUCC 11-0200) shared the same branch length with 72% ML support and grouped with the type strains of *A. didymospora* (MFLUCC 16-0412, MFLUCC 16-0416) with high support (100% ML, 100% MP, 1.00 PP) in Anastomitrabeculiaceae.

Analysis 2: Novel generated taxa in Parabambusicolaceae were analyzed with other representative genera in Parabambusicolaceae and other related families, including Bambusicolaceae, Dictyosporiaceae, Didymosphaeriaceae, Lentitheciaceae, Macrodiplodiopsidaceae, Sulcatisporaceae, and Trematosphaeriaceae based on a combined ITS, LSU, SSU, and tef1-α sequence dataset. The data matrix comprised 65 taxa with Melanomma pulvis-pyrius CBS 124080 (Melanommataceae) being the outgroup taxon. The dataset consists of 3828 total characters, including gaps (ITS: 1–601 bp, LSU: 602–1487 bp, SSU: 1488–2891 bp, *tef1-α*: 2892–3828 bp). The best scoring RAxML tree is presented in Figure 2 with a final ML optimization likelihood value of −25582.174441 (ln). RAxML analysis yielded 1375 distinct alignment patterns, and the proportion of gaps and completely undetermined characters in this alignment was 27.28%. The gamma distribution shape parameter alpha = 0.193883 and the Tree-Length = 2.813374. Estimated base frequencies were as follows: A = 0.236013, C = 0.254062, G = 0.272236, T = 0.237689, with substitution rates AC = 1.230009, AG = 2.581111, AT = 1.427698, CG = 1.028329, CT = 5.964523, GT = 1.000000. Bayesian analysis yielded 20,001 trees from two million runs, of which 15001 were sampled. Bayesian posterior probabilities (BYPP) from MCMC were evaluated with the final average standard deviation of split frequencies = 0.005045.

The phylogenetic results based on maximum likelihood (ML) and Bayesian inference analyses (Figure 2) showed overall similar tree topologies. A novel genus, *Paramultiseptospora* formed a stable subclade, clustered with the genera *Multiseptospora*, *Neomultiseptospora*, and *Scolecohyalosporium* with low support. These four genera formed a well-resolved clade (100% ML, 1.00 PP) within Parabambusicolaceae. 

Analysis 3: A new species, *Pyrenochaetopsis yunnanensis,* was analyzed with taxa in Pyrenochaetopsidaceae based on a combined LSU, ITS, *rpb2* and *tub2* DNA sequence dataset. The data matrix comprised 25 taxa, with *Neopyrenochaetopsis hominis* CBS 143033 being the outgroup taxon. The dataset consists of 2819 total characters, including gaps (LSU: 1–908 bp, ITS: 909–1468 bp, *rpb2*: 1469–2437 bp, *tub2*: 2438–2819 bp). The best scoring RAxML tree is presented in Figure 3 with a final ML optimization likelihood value of -12390.524384 (ln). RAxML analysis yielded 814 distinct alignment patterns, and the proportion of gaps and completely undetermined characters in this alignment was 7.92%. The proportion of invariable sites I = 0.518277 and the gamma distribution shape parameter alpha = 0.456186. The Tree-Length = 2.561446 with estimated base frequencies were as follows: A = 0.244976, C = 0.246101, G = 0.270141, T = 0.238781, and substitution rates AC = 2.182690, AG = 6.088919, AT = 2.687456, CG = 1.526108, CT = 12.069267, GT = 1.000000. The maximum parsimonious dataset consisted of 2819 characters, with 2067 characters being constant (proportion = 0.733239), 267 variable characters being parsimony-uninformative and 485 characters being parsimony-informative. The parsimonious analysis yielded eight parsimonious trees, of which the first parsimonious tree was selected as the best tree for the Kishino–Hasegawa test (TL = 1836, CI = 0.583, RI = 0.594, RC = 0.346, HI = 0.417). Bayesian analysis yielded 10,001 trees from one million runs, of which 7501 were sampled. Bayesian posterior probabilities (BYPP) from MCMC were evaluated with the final average standard deviation of split frequencies = 0.006366. 

Phylograms generated from maximum likelihood (ML), maximum parsimony (MP) and Bayesian inference analyses (Figure 3) were overall similar tree topologies. A new species, *Pyrenochaetopsis yunnanensis* (KUMCC 21-0843), has a close relationship with *P. terricola* (HGUP 1802) with high support (100% ML, 100% MP, 1.00 PP) and formed a well-resolved clade basal on *P. confluens* (CBS 142459), *P. decipiens* (CBS 343.85) and *P. indica* (CBS 124454) within Pyrenochaetopsidaceae.

Analysis 4: A new species, *Tetraploa bambusae*, was analyzed with other representative taxa in Tetraploasphaeriaceae based on a combined LSU, ITS, SSU, *tub2* and *tef1-α* DNA sequence dataset. The data matrix comprised 71 taxa, with *Muritestudina chiangraiensis* (MFLUCC 17-2551) being the outgroup taxon. The dataset consists of 3397 total characters, including gaps (LSU: 1–853 bp, ITS: 854–1427 bp, SSU: 1428–2421 bp, *tub2*: 2422–3078 bp, *tef1-α*: 3079–3397 bp). The best scoring RAxML tree is presented in Figure 4 with a final ML optimization likelihood value of −18736.220881 (ln). RAxML analysis yielded 1138 distinct alignment patterns, and the proportion of gaps and completely undetermined characters in this alignment was 27.96%. The proportion of invariable sites I = 0.573772 and the gamma distribution shape parameter alpha = 0.671292. The Tree-Length = 3.408354, and the estimated base frequencies were as follows: A = 0.239956, C = 0.252519, G = 0.274658, T = 0.232866, and substitution rates AC = 2.302918, AG = 3.658670, AT = 1.691247, CG = 1.425389, CT = 8.553285, GT = 1.000000. The maximum parsimonious dataset consisted of 3397 characters, with 2452 characters being constant (proportion = 0.721813), 177 variable characters being parsimony-uninformative and 768 characters being parsimony-informative. The parsimonious analysis yielded 1000 parsimonious trees, of which the first parsimonious tree was selected as the best tree for the Kishino–Hasegawa test (TL = 2787, CI = 0.532, RI = 0.832, RC = 0.443, HI = 0.468). Bayesian analysis yielded 10,001 trees from one million runs, of which 7501 were sampled. Bayesian posterior probabilities (BYPP) from MCMC were evaluated with the final average standard deviation of split frequencies = 0.007021. 

Phylograms generated from maximum likelihood (ML), maximum parsimony (MP) and Bayesian inference analyses (Figure 4) were overall similar tree topologies. A new species, Tetraploa bambusae (KUMCC 21-0844) formed a low support subclade (63% ML, 37% MP, 0.83 PP) with *Tetraploa* sp. (KT 1684) and clustered with *T. endophytica* (CBS 147114) and *T. obpyriformis* (KUMCC 21-0011) with high support (99% ML, 98% MP, 1.00 PP). 

### 3.2. Taxonomy

**Anastomitrabeculiaceae** Bhunjun, Phukhams. and K.D. Hyde

Bhunjun et al. [71] introduced the novel family Anastomitrabeculiaceae to accommodate a monotypic genus *Anastomitrabeculia* based on morphological characteristics and phylogenetic analyses of a combined LSU, SSU and *tef1-a* dataset coupled with divergence time estimates using molecular clock methodologies. The novel taxa were isolated from bamboo culms submerged in freshwater in southern Thailand. The genus is characterized by gregarious, uni-loculate, globose to subglobose, coriaceous ascomata, immersed under a clypeus to semi-immersed, with short, carbonaceous ostiolar neck, bitunicate, fissitunicate, cylindric-clavate asci, embedded in a hyaline, trabeculate pseudoparaphyses, and hyaline, fusiform, septate ascospores with longitudinally striate wall ornamentation, surrounded by a distinct, mucilaginous sheath [71]. According to Bhunjun et al. [71], Anastomitrabeculiaceae has a close phylogenetic relationship with Halojulellaceae. However, Halojulellaceae can be distinguished from Anastomitrabeculiaceae in having cellular pseudoparaphyses and pigmented ascospores. In this study, we collected *Anastomitrabeculia didymospora* from bamboo branches in terrestrial habitats in northern Thailand reported for the first time.

***Anastomitrabeculia didymospora*** Bhunjun, Phukhams. and K.D. Hyde, in Bhunjun, Phukhamsakda, Jeewon, Promputtha and Hyde, Journal of Fungi 7(2, no. 94): 12 (2021)

Index Fungorum number: IF 556559, Figure 5

Holotype information: Thailand, Krabi Province (8.1° N, 98.9° E), on dead bamboo culms submerged in freshwater, 15 December 2015, C. Phukhamsakda, KR001 (MFLU 20-0694), ex-type living culture = MFLUCC 16-0412.

Saprobic on dead branches of bamboo, visible as raised, black spots, with spike-like on the host surface. Sexual morph: Ascomata 200–320 μm high, 580–730 μm diam (excluding neck), gregarious, scattered to clustered, immersed under the clypeus to erumpent through host tissue by an ostiolar neck, ampulliform to subconical or hemispherical, uni-loculate, dark brown to black. Ostiolar neck 70–170 μm high, 130–200 μm diam, black, short, central, carbonaceous, papillate, protruding through host tissue. Peridium 30–100 μm wide at the sides, 6–15 μm wide at the base, unequally thick, poorly developed at the base, composed of fungal tissues intermixed with host tissues, of dark brown to black pseudoparenchymatous cells, arranged in a *textura angularis*. Hamathecium 1–2 μm wide, composed of dense, septate, branched, anastomosed, trabeculate pseudoparaphyses, embedded in a gelatinous matrix. Asci (100–)120–140(–170) × 16–20(–24) μm ($\overset{¯}{x}$ = 131.6 × 19.2 μm, *n* = 20), eight-spored, bitunicate, fissitunicate, cylindric-clavate, with a short pedicel, apically rounded with an ocular chamber. Ascospores (22–)25–30 × (6–)8–10 μm ($\overset{¯}{x}$ = 27.1 × 8.2 μm, *n* = 20), overlapping 1–2-seriate, hyaline, fusiform, straight to curved, 1(–3)-septate, wider in the upper part, rough-walled, with longitudinal furrows on the surface, surrounded by a distinct mucilaginous sheath. Asexual morph: Undetermined.

Culture characteristics: Ascospores germinated on PDA within 12 h. Colonies on PDA reaching 29–33 mm diam after 2 weeks at room temperature (30–35 °C). Colonies medium dense, irregular in shape, flat to slightly raised, surface smooth with an undulate edge, floccose to fluffy; colonies from above white at the margin, pale gray at the middle, with white hyphal turfs at the center; from below white to cream at the margin, yellowish-brown at the middle, dark greenish-gray to black at the center, slightly radiating inwards colony; not producing pigmentation on PDA.

Material examined: Thailand, Chiang Rai Province, Phan District, Mae Yen Subdistrict, Pu Khang Waterfall, on dead branches of bamboo, 13 January 2011, N.N. Wijayawardene, RP0113 (MFLU 11-0233), living culture: MFLUCC 11-0197; Chiang Mai Province, Mae Rim District, Mae Sa Waterfall, on dead branches of bamboo, 12 March 2011, R. Phookamsak, RP0116 (MFLU11-0236), living culture: MFLUCC 11-0200.

Known distribution: Krabi Province, southern Thailand [71], Chiang Mai and Chiang Rai Provinces, northern Thailand (this study).

Known host and habitats: saprobic on bamboo in freshwater [71] and terrestrial environments (this study).

Notes: The nucleotide BLAST search of ITS, LSU and *tef1-α* sequences resulted in the two newly generated strains (MFLUCC 11-0197 and MFLUCC 11-0200) being similar to *Anastomitrabeculia didymospora* MFLU 20-0694 (100% similarity). A nucleotide pairwise comparison of ITS, LSU and *tef1-α* sequences also indicated that strains MFLUCC 11-0197 and MFLUCC 11-0200 are consistent (less than 1.5% different base pair) with *A. didymospora* MFLU 20-0694 (type strain). We, therefore, identified our strains as *A. didymospora*. Morphologically, the new collection (MFLU 11-0233) is slightly larger in ascomata, asci, and ascospores than those of the type of *A. didymospora* [71]. The differences in the size range may be affected by environmental factors. Bhunjun et al. [71] mentioned that *A. didymospora* (MFLU 20-0694) has one-septate ascospores; however, we found that the species has 1(–3)-septate ascospores in this study. The host preference of *A. didymospora* is currently restricted to bamboo. However, the species is reported from terrestrial habitats for the first time.

**Figure 5 jof-08-00630-f005:**
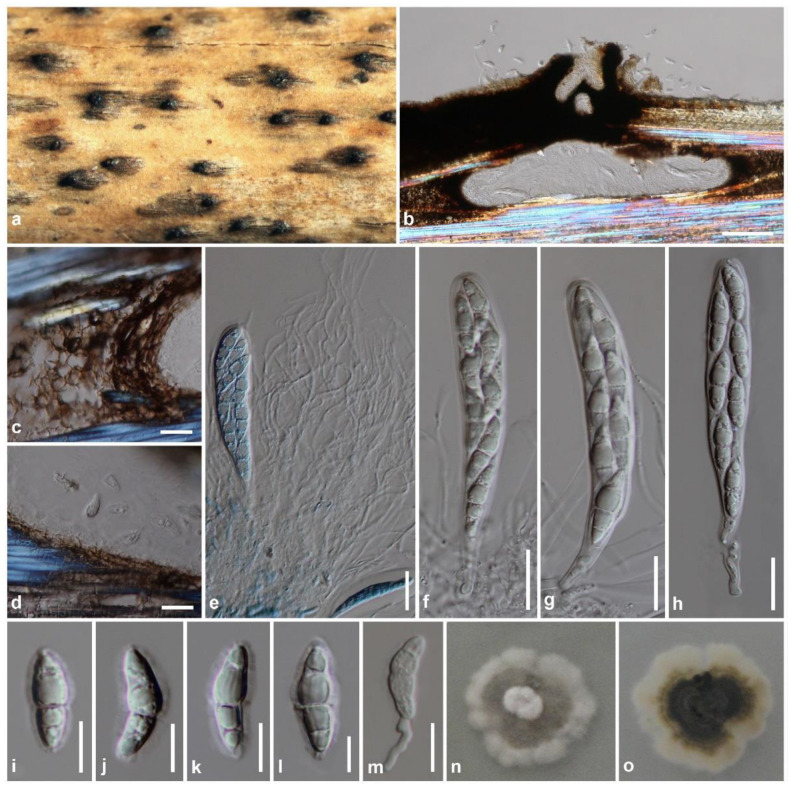
*Anastomitrabeculia didymospora* (MFLU 11-0233). (**a**) The appearance of ascomata on the host surface; (**b**) Vertical section of ascoma with ostiolar neck; (**c**,**d**) Peridium; (**e**) Pseudoparaphyses stained in cotton blue; (**f**–**h**) Asci; (**i**–**l**) Ascospores; (**m**) Germinating ascospore; (**n**,**o**) Culture characteristics on PDA after two weeks ((**n**) = from above, (**o**) = from below). Scale bars: (**b**) = 100 μm, (**c**–**h**) = 20 μm, (**i**–**m**) = 10 μm.

**Parabambusicolaceae** Kaz. Tanaka and K. Hiray.

Parabambusicolaceae was introduced by Tanaka et al. [27] to accommodate the genera *Aquastroma*, *Multiseptospora*, *Parabambusicola,* and the other two “*Monodictys* sp.”. Later, a monotypic genus *Multilocularia* was included in this family by Li et al. [100], while Wanasinghe et al. [101] and Phukhamsakda et al. [102] addressed both sexual and coelomycetous asexual species of *Neoaquastroma* in this family. Phukhamsakda et al. [102] also included *Pseudomonodictys* in Parabambusicolaceae. Subsequently, many genera were introduced in this family, including *Lonicericola*, *Neomultiseptospora*, *Paramonodictys*, *Paratrimmatostroma*, and *Scolecohyalosporium* [63,103,104]. Presently, 11 genera are accepted in this family based solely on the morpho-molecular approach. We follow the latest treatment of Xie et al. [104] and introduce the new genus *Paramultiseptospora* to accommodate a single species *P. bambusae* sp. nov. in this study.

***Paramultiseptospora*** Phookamsak, H.B. Jiang and Chomnunti, gen. nov.

Index Fungorum number: IF 554966

Etymology: Referring to relations with phylogenetically close genus *Multiseptospora*. Saprobic on dead stems of bamboo. Sexual morph: Ascomata gregarious, scattered to clustered, immersed in dark brown longitudinal clypeus, visible as raised, becoming superficial, lying along the host surface, uni-loculate, hemispherical to flattened ellipsoidal, or quadrilateral, glabrous, with apapillate ostiole. Peridium thin- to thick-walled, slightly thick at the sides, thinner at the apex, poorly developed at the base, composed of several layers of brown to dark brown, pseudoparenchymatous cells, paler brown to hyaline toward the inner layers, arranged in a *textura angularis*, outer layers intermixed with host tissues. Hamathecium composed of dense, branched, septate cellular pseudoparaphyses, anastomosed above the asci, embedded in a hyaline gelatinous matrix. Asci eight-spored, bitunicate, fissitunicate, cylindric-clavate to clavate, shortly pedicellate, apically rounded, with a well-developed ocular chamber. Ascospores overlapping one to three-seriate, hyaline, fusiform to oblong, with rounded ends, septate, constricted at the septa, smooth-walled, surrounded by a thick, mucilaginous sheath, with small guttules. Asexual morph: Undetermined. 

Type species: *Paramultiseptospora bambusae* Phookamsak and H.B. Jiang, sp. nov.

Notes: A monotypic genus *Paramultiseptospora* is introduced herein due to the differences in morphological characteristics with the other related genera (viz. *Multiseptospora*, *Neomultiseptospora* and *Scolecohyalosporium*), although the phylogenetic affinity of the genus does not support in this study. *Paramultiseptospora* formed a stable clade, closely related to *Multiseptospora* and *Scolecohyalosporium* in both BI and ML analyses and clustered with *Neomultiseptospora*. These four genera formed a well-resolved clade (100% ML, 1.00 PP; Figure 2) within Parabambusicolaceae. *Paramultiseptospora* can be easily distinguished from *Multiseptospora* and *Scolecohyalosporium* in having hemispherical to flattened ellipsoidal, glabrous, ascomata, immersed in longitudinal clypeus, visible as raised, lying along the host surface, cylindric-clavate to clavate asci with short pedicel and fusiform to oblong ascospores, with rounded ends. Meanwhile, *Multiseptospora* has globose to subglobose ascomata, immersed in the host, covered by dark, hair-like hyphae, broadly cylindrical, subsessile asci, and fusiform to vermiform ascospore, with acute ends [33]. *Scolecohyalosporium* is different in having conical to ovoid, black, rough-walled ascomata, erumpent to superficial on the host, long cylindrical asci, with short pedicel and filiform ascospores [104]. *Paramultiseptospora* morphologically resembles *Neomultiseptospora* in having hemispherical to subconical, glabrous ascomata, immersed in the host, with apapillate ostiole, clavate asci, with short pedicel and fusiform or oblong, septate ascospores, surrounded by a thick mucilaginous sheath. However, these two genera are slightly different in the characteristics of ascomata on the host. *Paramultiseptospora* formed gregarious, scattered to clustered ascomata, immersed in dark brown longitudinal clypeus, lying along the host surface whereas *Neomultiseptospora* formed solitary, scattered, immersed, visible as raised, black dome-shaped on the host surface [104]. Phylogenetically, *Paramultiseptospora* always formed a separate branch from *Neomultiseptospora*. Therefore, we consider *Paramultiseptospora* as a distinct genus with *Neomultiseptospora* based on morphology coupled with the phylogenetic relationship.

***Paramultiseptospora bambusae*** Phookamsak and H.B. Jiang, sp. nov.

Index Fungorum number: IF 554968, Figure 6

Etymology: Referring to the host, bamboo, of on which the species was collected.

Holotype: KUN-HKAS 122241

Saprobic on a dead stem of bamboo. Sexual morph: Ascomata 115–150 μm high, 340–470 μm diam, gregarious, scattered to clustered, immersed in dark brown longitudinal clypeus, visible as raised, becoming superficial, lying along the host surface, uni-loculate, hemispherical to flattened ellipsoidal, or quadrilateral, glabrous, indistinct apapillate ostiole. Peridium 30–90 μm wide at the sides toward the apex, 10–25 μm wide at the base, thin- to thick-walled, slightly thick at the sides, thinner at the apex, poorly developed at the base, composed of several layers of brown to dark brown, pseudoparenchymatous cells, paler brown to hyaline toward the inner layers, arranged in a *textura angularis*, outer layers intermixed with host tissues. Hamathecium composed of dense, 1–2.5 μm wide, branched, septate cellular pseudoparaphyses, anastomosed above the asci, embedded in a hyaline gelatinous matrix. Asci (70–)75–90(–95) × 16–20(–22) μm ($\overset{¯}{x}$ = 83.3 × 19.6 μm, *n* = 30), eight-spored, bitunicate, fissitunicate, cylindric-clavate to clavate, shortly pedicellate, apically rounded, with a well-developed ocular chamber. Ascospores (23–)25–28(–29) × 5–8 μm ($\overset{¯}{x}$ = 25.8 × 6.6 μm, *n* = 30), overlapping one to three-seriate, hyaline, fusiform to oblong, with rounded ends, narrower toward the end cells, enlarged at the third cell from above, slightly curved, six-septate, constricted at the septa, smooth-walled, surrounded by a thick, mucilaginous sheath, with small guttules. Asexual morph: Undetermined

Material examined: China, Yunnan Province, Honghe Autonomous Prefecture, Honghe County, Honghe Hani Rice Terraces (23°5′35″ N, 102°46′47″ E, 1432 + 6 msl), on dead stem of bamboo, 26 January 2021, R. Phookamsak, BN09F (KUN-HKAS 122241, holotype), ex-type strain: KUN-HKAS 122241A. Notes: DNA was extracted from fruit bodies.

Known distribution: Yunnan Province, China.

Known host and habitats: saprobic on a stem of bamboo in a terrestrial environment.

Notes: The nucleotide BLAST search of ITS sequence indicated that *Paramultiseptospora bambusae* (KUN-HKAS 122241A) has the closest similarity with *Multiseptospora thailandica* strain MFLUCC 11-0183 (ex-type strain) with 95.58% similarity (Identities = 432/452, with no gap), strains MFLUCC 11-0204 and MFLUCC 12-0006 with 95.48% similarity (Identities = 444/465, with no gap) and is similar to “Pleosporales sp. strain 1192” (95.54% similarity, Identities = 407/426, with no gap). *Paramultiseptospora bambusae* (KUN-HKAS 122241A) also matches with *Neomultiseptospora yunnanensis* strain KUMCC 21-0411 (ex-type strain) with 92.84% similarity (Identities = 428/461, with no gap) and *Scolecohyalosporium submersum* strain KUMCC 21-0412 (ex-type strain) with 92.57% similarity (Identities = 436/471, with two gaps). The nucleotide BLAST search of LSU sequence indicated that *P. bambusae* (KUN-HKAS 122241A) is similar to *S. submersum* strains KUMCC 21-0412, KUMCC 21-0413 and KUN-HKAS 122242 with 98.57% similarity (Identities = 830/842, with two gaps), similar to *M. thailandica* strain MFLUCC 12-0006 (98.56% similarity, Identities = 830/843, with four gaps) and strain MFLUCC 11-0204 (98.46% similarity, Identities = 821/833, with three gaps), and is similar to *N. yunnanensis* strain KUMCC 21-0411 (97.59% similarity, Identities = 811/831, with one gap) and strain KUN-HKAS 122240 (97.23% similarity, Identities = 808/831, with one gap). 

Based on a nucleotide pairwise comparison, *Paramultiseptospora bambusae* (KUN-HKAS 122241A) differs from *Multiseptospora thailandica* (MFLUCC 11-0183, ex-type strain) in 88/570 bp of ITS (15.44%), 13/765 bp of LSU (1.7%), and 28/645 bp of *tef1*-α (4.34%). *Paramultiseptospora bambusae* (KUN-HKAS 122241A) differs from *Scolecohyalosporium submersum* (KUMCC 21-0412) in 85/595 bp of ITS (14.28%), 12/842 bp of LSU (1.42%), and 39/921 bp of *tef1-α* (4.23%). The species is also different from *Neomultiseptospora yunnanensis* strain KUMCC 21-0411 (ex-type strain) in 102/606 bp of ITS (16.83%), 21/832 bp of LSU (2.52%), and 52/979 bp of *tef1-α* (5.31%). *Paramultiseptospora bambusae* is morphologically similar to *N. yunnanensis* but differs in having fusiform to oblong, six-septate ascospores with rounded ends, narrower toward the end cells, and constricted at the septa, whereas *N. yunnanensis* has fusiform to ellipsoidal, or oblong, (four to) five-septate ascospores, with rounded ends, slightly constricted at the central septum, which are less constricted at the other septa [104].

**Figure 6 jof-08-00630-f006:**
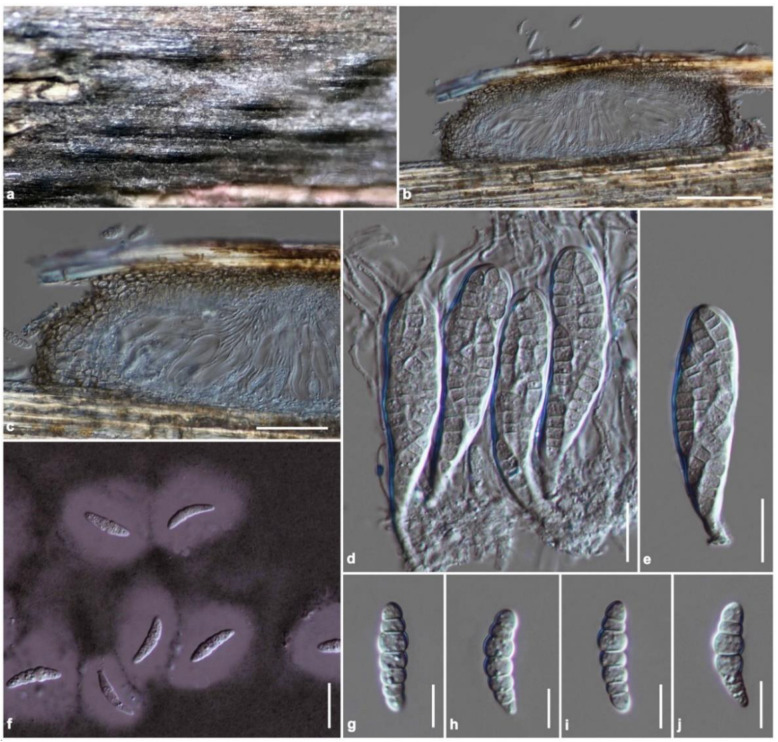
*Paramultiseptospora bambusae* (KUN-HKAS 122241, holotype). (**a**) The appearance of ascomata on host substrate; (**b**) Vertical section of ascoma; (**c**) Peridium; (**d**) Asci embedded in cellular pseudoparaphyses; (**e**) Ascus; (**f**) Ascospores stained with Indian ink showing a thick mucilaginous sheath surrounded ascospores; (**g**–**j**) Ascospores. Scale bars: (**b**) = 100 μm, (**c**) = 50 μm, (**d**–**f**) = 20 μm, (**g**–**j**) = 10 μm.

**Pyrenochaetopsidaceae** Valenzuela-Lopez et al.

Pyrenochaetopsidaceae was introduced by Valenzuela Lopez et al. [105] to accommodate the asexual genera *Pyrenochaetopsis* (type genus), *Neopyrenochaetopsis* and X*enopyrenochaetopsis*. The family is characterized by pycnidial, pale brown to brown, solitary or confluent, glabrous or setose, subglobose to ovoid conidiomata, with apapillate or papillate ostiole, acropleurogenous conidiophores, phialidic, hyaline, discrete or integrated, septate conidiogenous cells, and aseptate, hyaline, smooth- and thin-walled, ovoid, cylindrical to allantoid ascospores [105]. Mapook et al. [106] introduced a novel species, *Pyrenochaetopsis chromolaenae* collected on *Chromolaena odorata* in Thailand and reported the sexual morph of *Pyrenochaetopsis* for the first time. The sexual morph is characterized by brown to dark brown solitary or scattered, globose ascomata, superficial on the host, with short papillate ostiole, with reddish-brown setae covering the papilla, thin-walled peridium, fissitunicate, cylindric-clavate asci, with a short, bulbous pedicel, and hyaline to pale brown or yellowish-brown, cylindrical to broadly fusiform, three to four-septate ascospores [106]. Species in Pyrenochaetopsidaceae have been isolated from various substrates as saprobes and also opportunistic pathogens on humans as well as on cysts of plant-parasitic nematodes. [81,105,106].

*Pyrenochaetopsis* was treated as the generic type of Pyrenochaetopsidaceae and is typified by *P. leptospora*. The genus was introduced by de Gruyter et al. [107] to accommodate phoma-like taxa. Recently, 19 species are accepted in this genus [108]. In this study, we introduce a holomorph species, *P. yunnanensis,* which occurred on bamboo in Yunnan, China.

***Pyrenochaetopsis yunnanensis*** C.F. Liao, H.B. Jiang and Phookamsak, sp. nov.

Index Fungorum number: IF 554979, Figure 7

Etymology: Referring to the locality, Yunnan Province of China, of which the species was collected.

Holotype: KUN-HKAS 123172

Saprobic on dead stem of bamboo. Sexual morph: Ascomata 190–280 μm high, 270–460 μm diam, gregarious, scattered to clustered, immersed to semi-immersed under the clypeus, visible as raised, black, shiny, rough on host surface, uni- to tri-loculate, subglobose to subconical, or quadrilateral, glabrous, ostiole central with minute papillate, protruding through host tissue. Peridium 30–80 μm wide at sides toward the apex, 10–30 μm wide at the base, unequally thickness, poorly developed at the base, composed of several layers of dark brown pseudoparenchymatous cells, arranged in *textura angularis* to *textura prismatica*, outer layers intermixed with host cortex. Hamathecium composed of dense, 1.5–2.5 μm wide, filamentous, branched, septate, cellular pseudoparaphyses, anastomosed above the asci, embedded in a hyaline gelatinous matrix. Asci (58–)70–95(–110) × 11–14(–15.5) μm ($\overset{¯}{x}$ = 80.2 × 12.8 μm, *n* = 30), eight-spored, bitunicate, fissitunicate, clavate, with short pedicel, apically rounded, with a well-developed ocular chamber. Ascospores (18–)20–25(–28) × (4.5–)5–6.5(–8) μm ($\overset{¯}{x}$ = 23.4 × 5.8 μm, *n* = 30), overlapping one to three-seriate, hyaline, fusiform, with acute ends, slightly curved, one to three-septate, slightly constricted at the central septum, not constricted at the other septa, smooth-walled, lacking mucilaginous sheath. Asexual morph: Coelomycetous, sporulated on PDA after two months at room temperature (15–20 °C), visible as black dots, superficial or immersed in PDA. Conidiomata 50–100 μm high, 68–105 μm diam, pycnidial, black, solitary, or in a small group, immersed to superficial, globose to subglobose, uni- to multi-loculate, setose, with dark brown, septate setae (35–75 × 2–4 μm, *n* = 20), ostiole central, with pore-like opening or pimple-like. Peridial wall 7.5–20 μm wide, equally thin-walled, composed of one to two layers, of brown to dark brown pseudoparenchymatous cells, arranged in *textura angularis* to *textura prismatica*, or *textura globulosa*. Conidiophores reduced to conidiogenous cells. Conidiogenous cells 2–4 × (1.8–)2–3.5 μm ($\overset{¯}{x}$ = 3.5 × 2.8 μm, *n* = 30) phialidic, hyaline, discrete, aseptate, arising from the inner cavity of the pycnidial wall, difficult to distinguish from the pycnidial wall. Conidia (2.8–)3–4 × 1–2(–2.5) μm ($\overset{¯}{x}$ = 3.4 × 1.9 μm, *n* = 50) hyaline, subglobose to oblong, aseptate, with small guttules.

Culture characteristics: Ascospores germinated on PDA within 24 h. Colonies on PDA reach 25–28 mm diam after one week at room temperature (15–20 °C). Colonies medium dense, irregular in shape, flat to slightly raised, surface smooth with a lobate edge, floccose to cottony; colony from above pale gray to white-gray at the margin, white at the middle toward the center, sectored; from below white at the margin, pale yellowish-brown at middle toward the center; not producing pigmentation on PDA. Sporulation on PDA after two months.

Material examined: China, Yunnan Province, Honghe Autonomous Prefecture, Honghe County, Honghe Hani Rice Terraces (23°5′35″ N, 102°46′47″ E, 1432 + 6 msl), on dead stem of bamboo in a terrestrial environment, 26 January 2021, R. Phookamsak, BN09F (KUN-HKAS 123172, holotype), ex-type living culture: KUMCC 21-0843.

Known distribution: Yunnan Province, China.

Known host and habitats: saprobic on a stem of bamboo in a terrestrial environment.

Notes: The nucleotide BLAST search of ITS sequence indicated that *Pyrenochaetopsis yunnanensis* (KUMCC 21-0843) is similar to *Leptosphaeria* sp. (isolate NTOU5272 and R10) with 100% similarity (Identities = 521/521 and 490/490, with no gap), it is similar to *Pyrenochaetopsis terricola* strain HGUP1802 (ex-type strain) with 99.41% similarity (Identities = 506/509, with two gaps), and it is also identical to *Dokmaia* sp. isolate C126 (99.79% similarity, Identities = 485/486, with one gap). The nucleotide BLAST search of *rpb2* sequence indicated that *P. yunnanensis* (KUMCC 21-0843) is similar to *P. terricola* strain HGUP1802 with 94.71% similarity (Identities = 948/1001, with no gap), and the nucleotide BLAST search of *tub2* sequence also showed that *P. yunnanensis* (KUMCC 21-0843) is identical to *P. terricola* strain HGUP1802 with 98.61% similarity (Identities = 354/359, with three gaps) and is identical to *P. sinensis* strain LC12199 with 92.98% similarity (Identities = 265/285, with six gaps). 

Based on a nucleotide pairwise comparison, *Pyrenochaetopsis yunnanensis* (KUMCC 21-0843) is consistent with *P. terricola* strain HGUP1802 in LSU nucleotide pairwise comparison but differs from *P. terricola* in 11/519 bp of ITS (2.12%), 53/1002 bp of *rpb2* (5.29%), and 7/363 bp of *tub2* (1.93%). *Pyrenochaetopsis yunnanensis* (KUMCC 21-0843) grouped with *P. terricola* strain HGUP1802 with high support (100% ML, 100% MP, 1.00 PP; Figure 3) in the present study. *Pyrenochaetopsis yunnanensis* (KUMCC 21-0843) morphological resembles *P. terricola* but the conidial size is slightly longer than *P. terricola* (2–3 × 1–2 μm) [109]. Wang et al. [109] isolated *P. terricola* from the soil in Guizhou Province, China and determined only the asexual morph sporulated on OA, while our novel species was found as a saprobe on bamboo and both sexual and asexual morph.

**Tetraplosphaeriaceae** Kaz. Tanaka and K. Hiray.

Tetraplosphaeriaceae was introduced by Tanaka et al. [12] to accommodate genera that mostly occurred on bamboo. Five genera that formed tetraploa-like asexual morph were initially introduced to this family, including *Polyplosphaeria*, *Pseudotetraploa*, *Quadricrura*, *Tetraplosphaeria* (generic type), and *Triplosphaeria* [12]. Later, *Tetraplosphaeria* was treated as a synonym of *Tetraploa* [72,110]. Recently, nine genera were accepted in this family viz. *Aquatisphaeria*, *Byssolophis*, *Ernakulamia*, *Polyplosphaeria*, *Pseudotetraploa*, *Quadricrura*, *Shrungabeeja*, *Tetraploa* (= *Tetraplosphaeria*), and *Triplosphaeria* [41,81,111,112,113,114]. Most species in Tetraplosphaeriaceae were reported as saprobes on bamboo, but some species were isolated from soil and water [81].

*Tetraploa* (= *Tetraplosphaeria*), generic type of Tetraplosphaeriaceae, was introduced by Berkeley and Broome [115] with *T. aristata* as the type species. The asexual morph of *Tetraploa* is characterized by lacking conidiophores, monoblastic conidiogenous cells, and brown, short-cylindrical, verrucose conidia, composed of four columns with four setose appendages at the apex [12,81]. The sexual morph is characterized by scattered to gregarious, immersed to erumpent, globose to subglobose, glabrous ascomata, with short-papillate to cylindrical ostiole, fissitunicate, cylindrical to clavate, short-pedicellate asci, and hyaline, narrowly fusiform, septate, smooth-walled ascospores, surrounded by a mucilaginous appendage-like sheath [12,81]. Species in *Tetraploa* mostly occurred on bamboos and other herbaceous plants or rotten wood as well as isolated from soil or raindrops [81]. In this study, the novel species, *T. bambusae*, isolated from bamboo in Yunnan, China is introduced based on morphological characteristics and multigene phylogenetic analyses.

**Figure 7 jof-08-00630-f007:**
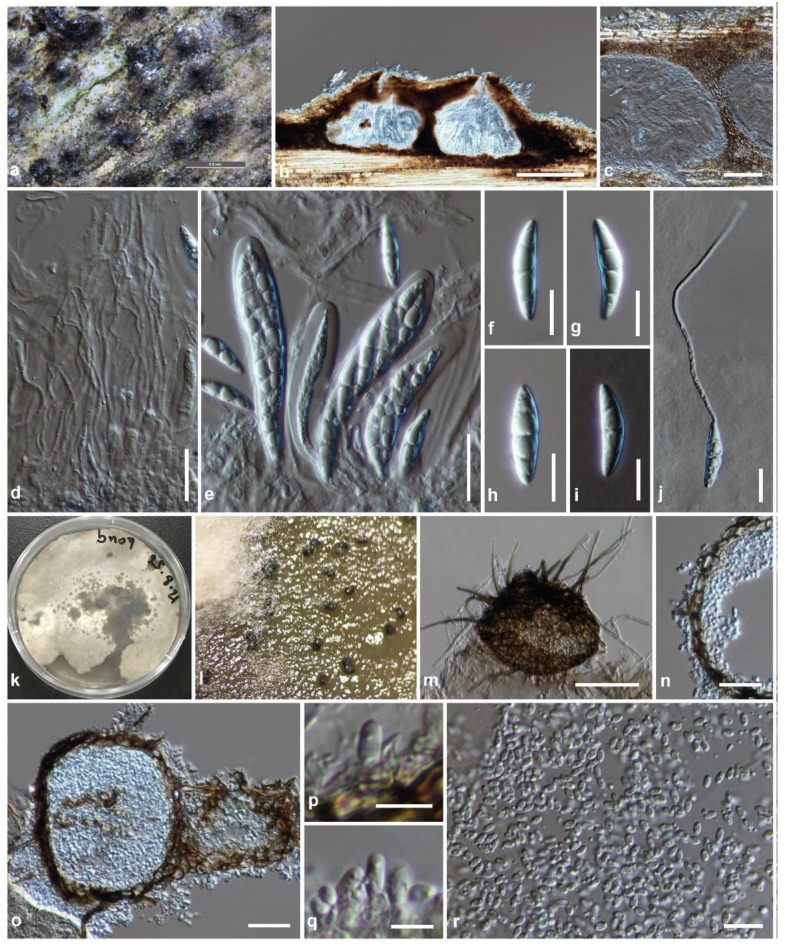
*Pyrenochaetopsis yunnanensis* (KUN-HKAS 123172, holotype). (**a**) The appearance of ascomata on host substrate; (**b**) Vertical section of ascomata; (**c**) Peridium; (**d**) Pseudoparaphyses; (**e**) Asci; (**f**–**h**) Ascospores; (**i**) Ascospore stained with Indian ink; (**j**) Germinated ascospore; (**k**) Colony sporulated on PDA after two months; (**l**) Conidiomata immersed or superficial on PDA; (**m**) Squash mount of conidioma in water; (**n**) Section through pycnidial wall; (**o**) Vertical section of conidiomata; (**p**,**q**) Conidiogenous cells; (**r**) Conidia. Scale bars: (**b**) = 200 μm, (**c**,**m**) = 50 μm, (**d**,**e**,**n**,**o**) = 20 μm, (**f**–**i**,**j**,**r**) = 10 μm, (**p**,**q**) = 5 μm.

***Tetraploa bambusae*** Phookamsak and H.B. Jiang, sp. nov. 

Index Fungorum number: IF 554987, Figure 8

Etymology: Referring to the host, bamboo, of which the species was collected.

Holotype: KUN-HKAS 123174

Saprobic on dead twigs of bamboo. Sexual morph: Undetermined. Asexual morph: hyphomycetous. Colonies brown to brick orange, effuse to powdery, compact, with patch-like, superficial on the host substrate. Mycelia light brown to brown, branched, septate. Conidiophores up to 40–130 µm long, (1.5–)2–3.5 µm wide, macronematous, inconspicuous, light brown, branched, septate. Conidiogenous cells monoblastic, discrete or integrated, determinate, cylindrical. Conidia (21–)23–30(–33) × (17–)18–23(–26) µm ($\overset{¯}{x}$ = 24.1 × 19.8 µm, *n* = 30), muriform, obovoid to turbinate, with obtuse end, brown to dark brown, composed of four columns of cells, four-septate in each column, coarsely verruculose, with four apical appendages, sometimes, a small piece of the denticle remains attached to the base of the conidium. Appendages 15–40 µm long, 2.5–4.5 µm wide at the base, wider at the base, tapering toward the apex, divergent, brown, one to three-septate, straight or slightly flexuous, smooth-walled. 

Culture characteristics: Ascospores germinated on PDA within 24 h. Colonies on PDA reach 22–25 mm diam after two weeks at room temperature (15–20 °C). Colonies dense, irregular in shape, raised to umbonate, surface smooth with an undulate edge, velvety; colony from above white–gray to pale gray at the margin, gray at the middle toward the center; from below white to cream at the margin, orange–brown at the middle, brown to dark brown at the center, slightly radiated outwards colony with concentric rings; not producing pigmentation on PDA. 

Material examined: China, Yunnan Province, Kunming City, Kunming Institute of Botany, on dead twigs of bamboo, 31 January 2021, R. Phookamsak, KIB21-005 (KUN-HKAS 123174, holotype), ex-type living culture: KUMCC 21-0844.

Known distribution: Yunnan Province, China.

Known host and habitats: saprobic on twigs of bamboo in a terrestrial environment.

Notes: The nucleotide BLAST search of ITS sequence indicated that *Tetraploa bambusae* (KUMCC 21-0844) closest matches with “uncultured fungus” clone 035A11084, 109A74706, 036A17775, and 034A2039 with 96.30%, 96.05%, 95.80%, and 95.71% similarities, respectively. The species is similar to *Tetraplosphaeria* sp. strain WSF14_RG24_2 with 95.62% similarity (Identities = 502/525, with eight gaps), and it is similar to *Tetraploa yunnanensis* MFLUCC 19-0319 (ex-type strain) with 95.60% similarity (Identities = 521/545, with nine gaps). The nucleotide BLAST search of LSU sequence indicated that *T. bambusae* (KUMCC 21-0844) closest matches with *T. sasicola* strain MFLUCC 17-1387 with 99.88% similarity (Identities = 822/823, with no gap) and is similar to *Tetraploa obpyriformis* KUMCC 21-0011 with 99.64% similarity (Identities = 821/824, with one gap) and *Tetraploa* sp. KT 1684 with 99.51% similarity (Identities = 816/820, with one gap). The nucleotide BLAST search of *tub2* sequence also showed that *T. bambusae* (KUMCC 21-0844) closest matches with *Tetraplosphaeria sasicola* KT 563 with 88.51% similarity (Identities = 578/653, with 29 gaps) and is similar to *Tetraploa aristata* CBS 996.70 with 82.53% similarity (Identities = 529/641, with 27 gaps) and *Tetraplosphaeria yakushimensis* KT 1906 with 81.58% similarity (Identities = 536/657, with 38 gaps). 

Phylogenetic analyses based on a combined LSU-ITS-SSU-*tub2*-*tef1-α* sequence dataset demonstrated that *Tetraploa bambusae* (KUMCC 21-0844) is sister to *Tetraploa* sp. KT 1684 and clustered with *T. endophytica* CBS 147114 and *T. obpyriformis* KUMCC 21-0011 with high support (99% ML, 98% MP, 1.00 PP; Figure 4). Based on a nucleotide pairwise comparison, *T. bambusae* (KUMCC 21-0844) is consistent with *Tetraploa* sp. KT 1684 in LSU nucleotide pairwise comparison (differs in 1 bp), but it could not be compared for the other informative gene regions (viz. ITS, *tub2*, and *tef1-α*) due to the lack of sequence data of *Tetraploa* sp. KT 1684. Tanaka et al. [12] included *Tetraploa* sp. KT 1684 in their analyses when they introduced the new family Tetraplosphaeriaceae; however, the morphological characteristics of *Tetraploa* sp. KT 1684 were not described. Thus, we could not compare the morphology of the novel species with *Tetraploa* sp. KT 1684, while *T. obpyriformis* KUMCC 21-0011 is an unpublished species. *Tetraploa endophytica* CBS 147114 was isolated from the roots of *Microthlaspi perfoliatum* (Brassicaceae) as an endophyte. The strain did not sporulate in any of the different culture media [116]. Therefore, the species also could not compare their morphology.

**Figure 8 jof-08-00630-f008:**
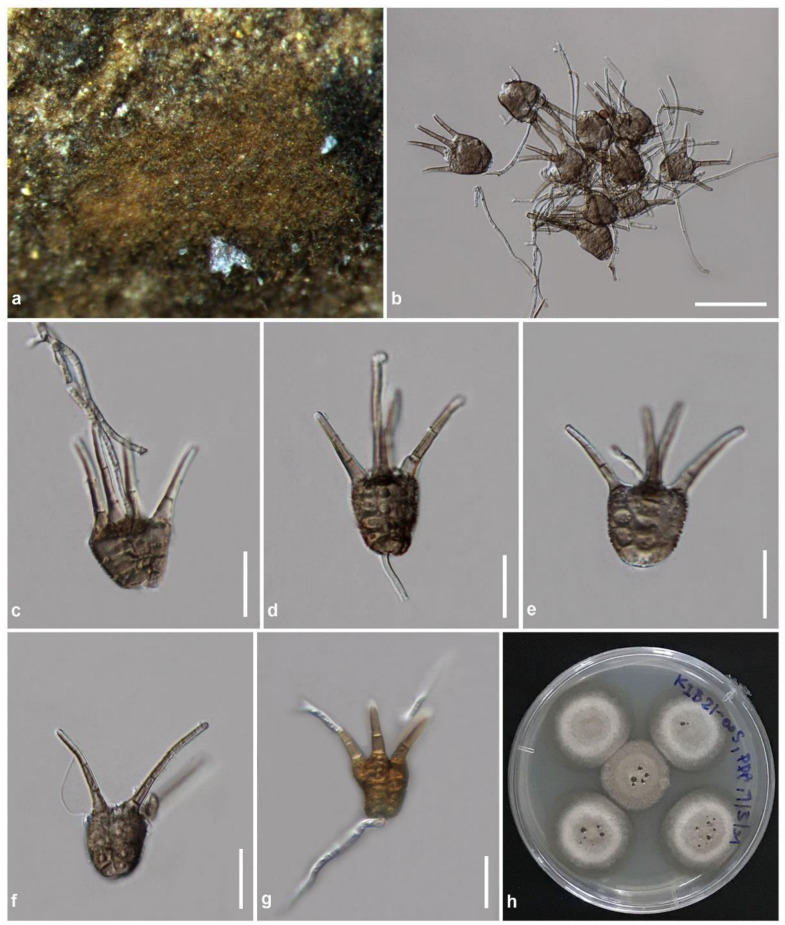
*Tetraploa bambusae* (KUN-HKAS 123174, holotype). (**a**) The appearance of colony on host substrate; (**b**) Conidial mass; (**c**,**d**) Conidia attached with conidiophores; (**e**,**f**) Conidia; (**g**) Germinated conidium; (**h**) Culture characteristics on PDA after one week. Scale bars: (**b**) = 50 μm, (**c**–**g**) = 20 μm.

## 4. Discussion and Conclusions

Bambusicolous fungi are highly diverse and distributed in various families within Pleosporales. Since 2015, over 85 bambusicolous species have been introduced in Pleosporales [3,27,33,39,41,45,46,49,50,52,55,59,63,68,71,74,76,77,78,79,80,100,104,117,118,119,120,121,122,123,124,125,126,127,128,129,130,131,132]. Even though novel taxa have been continuedly introduced in recent years, studies into the diversity of bambusicolous fungi correlating with specific bamboo genera are still limited due to the lack of host species identification. Jiang [133] reported that 48% of pleosporalean taxa were associated with bamboos in Thailand, and 39% were discovered in Yunnan, China. However, documented species were restricted to some parts of Thailand (mainly found in northern) and Yunnan Province (e.g., Honghe, Kunming, and Xishuangbanna) as well as some provinces of China (e.g., Guangdong and Sichuan) and Japan, whereas other regions that show a high species richness of bamboos have received less attention. Hence, we believe that a huge number of novel taxa occurring on bamboo are waiting for discovery in other regions. In this study, we collected the ascomycetes on bamboo in Honghe and Kunming (Yunnan Province, China) and also included the collections from Chiang Mai and Chiang Rai Provinces (Thailand), which were collected in 2011. Based on morphological characteristics and multigene phylogenetic analyses, four novel taxa collected from Yunnan, China are introduced, including *Paramultiseptospora bambusae* sp. et gen. nov., *Pyrenochaetopsis yunnanensis* sp. nov., and *Tetraploa bambusae* sp. nov., whereas collections from Thailand are identified as *Anastomitrabeculia didymospora* and reported in terrestrial habitats for the first time.

*Anastomitrabeculia didymospora* is a putative species accommodated in Anastomitrabeculiaceae. Bhunjun et al. [71] reported that the species that occurred on the bamboo host was submerged in freshwater. In this study, we also found the species occurring on a bamboo host in the terrestrial habitat near the waterfall. There are few studies concerning relationships between freshwater and terrestrial fungi [134,135,136,137]. Boonyuen et al. [136] mentioned that fungal species partially overlap between freshwater and terrestrial habitats, of which the submerged samples yielded the most fungal diversity. Boonyuen et al. [136] also suggested that the diversity of wood-inhabiting fungi depended on tree species, geography, and exposure period. There is no clear evidence to prove that terrestrial fungi will continue to thrive when submerged in water [137]. Kodseub et al. [137] attempted to investigate the differences in fungal communities that occurred in *Magnolia liliifera* wood from freshwater and terrestrial habitats. Kodseub et al. [137] mentioned that dominant fungi in the terrestrial environment were significantly different from fungi submerged in freshwater, and few species have been found in both freshwater and terrestrial habitats, suggesting that most fungi occurring on wood in terrestrial habitats did not thrive in freshwater habitats. According to Kodseub et al. [137], we hypothesized that *A. didymospora* is one of the few species that can survive in both freshwater and terrestrial habitats. The species may initially occur on a bamboo host in the terrestrial environment and continue to thrive in submerged freshwater.

Parabambusicolaceae shows to be heterogeneous, and it currently contains 12 genera, including the new genus introduced in this study. Even though most genera of Parabambusicolaceae contain a single species, they showed high genetic heterogeneity, which can be interpreted by their phylogenetic relationships. Most genera in Parabambusicolaceae are only represented by their sexual or asexual morph, except for *Neoaquastroma*. Hence, the morphology of some sexual and asexual genera could not be compared, which led to the generic status becoming questionable. More taxa sampling is required for a better understanding of each genus in Parabambusicolaceae.

*Pyrenochaetopsis* was introduced to accommodate phoma-like taxa that occurred on various host substrates [81,105,106,138]. The genus was previously treated in Cucurbitariaceae [72,107,139]. Later, Valenzuela-Lopez et al. [105] introduced the new family Pyrenochaetopsidaceae to accommodate this genus together with *Neopyrenochaetopsis* and *Xenopyrenochaetopsis*. Mapook et al. [106] determined the sexual morph of *Pyrenochaetopsis*, *P. chromolaenae*, for the first time. In the present study, the holomorph of *P. yunnanensis* sp. nov. is also determined. The sexual morph of *P. yunnanensis* can be distinguished from *P. chromolaenae* in having subglobose to subconical, or quadrilateral, glabrous ascomata and hyaline, fusiform, one to three-septate ascospores, whereas *P. chromolaenae* has globose ascomata with setose papilla and hyaline to pale brown or yellowish-brown, cylindrical to broadly fusiform, three to four-septate ascospores [106]. *Pyrenochaetopsis yunnanensis* is reported as a saprobe on bamboo host in Yunnan Province, China for the first time. Species of *Pyrenochaetopsis* are well-studied based on molecular analyses coupled with morphological characteristics of their asexual morph. Nevertheless, the sexual morph of this genus is still rarely detected. 

In the present study, multigene phylogenetic analyses demonstrated that taxa in *Tetraploa* could be separated into two subclades. The main subclade (including the type species) comprises *T. aristata* (type species), *T. bambusae* sp. nov., *T. dwibahubeeja*, *T. endophytica*, *T. obpyriformis*, *T. pseudoaristata*, *T. puzheheiensis*, *T. sasicola*, *T. thrayabahubeeja*, *T. yakushimensis*, *T. yunnanensis,* and *Tetraploa* spp. (CY 112, KT 1684). These species formed a well-resolved subclade within Tetraplosphaeriaceae (Figure 4). The second subclade comprises *T. aquatica*, *T. cylindrica*, *T. nagasakiensis* and *Tetraploa* sp. (KT 2578). *Tetraploa aquatica*, *T. cylindrica,* and *T. nagasakiensis* formed a well-resolved subclade, clustered with *Tetraploa* sp. KT 2578 with low support and constituted independently basal to the main subclade. Multigene phylogenetic analyses showed that *T. aquatica*, *T. cylindrica,* and *T. nagasakiensis* may be distinct genera with *Tetraploa*, but generic clarification insight into the morphology-based taxonomy is needed in the future study. *Tetraploa sasicola* (KT 563, ex-type strain) also formed a separate branch with *T. sasicola* (FU31019) in this study and also concurred with Liao et al. [140]. *Tetraploa sasicola* strain FU31019 may not be conspecific with *T. sasicola* (KT 563) pending further study.

## Figures and Tables

**Figure 1 jof-08-00630-f001:**
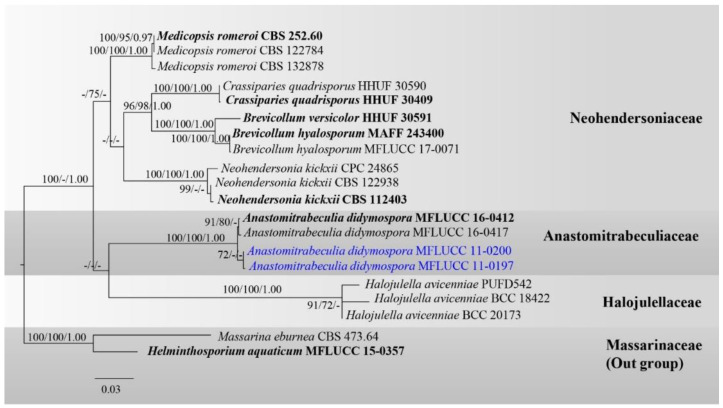
Phylogram generated from RAxML analysis of a concatenated LSU-SSU-ITS-*tef1-α* sequence dataset to represent the phylogenetic relationships of taxa in Anastomitrabeculiaceae, Halojulellaceae, and Neohendersoniaceae. Bootstrap support values for ML and MP equal to or greater than 70% and the Bayesian posterior probabilities equal to or higher than 0.95 PP are indicated above the nodes as ML/MP/PP. Support values lower than 70% ML/MP and 0.95 PP are indicated by a hyphen (-). Ex-type strains are in bold and the newly generated sequences are indicated in blue.

**Figure 2 jof-08-00630-f002:**
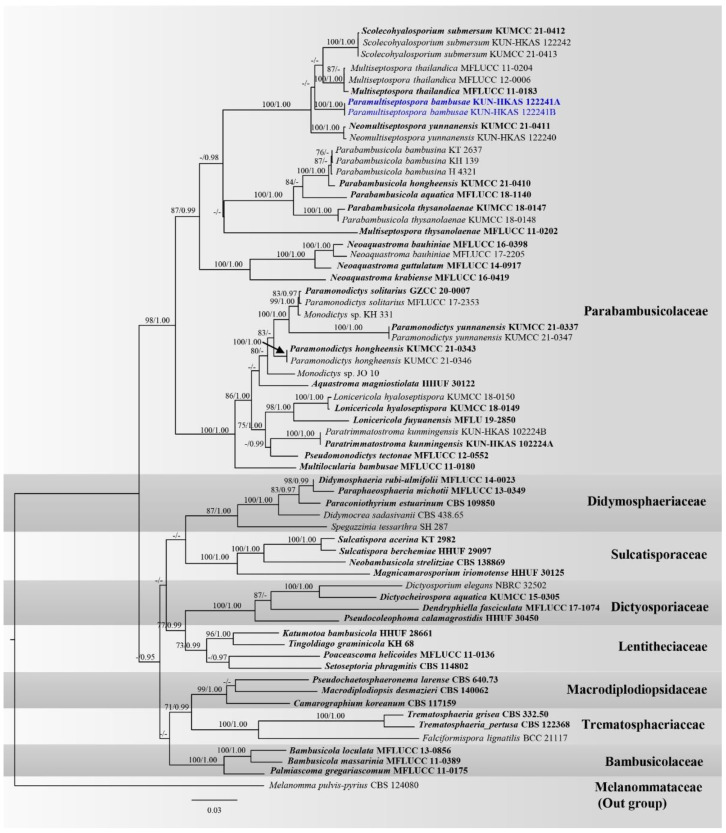
Phylogram generated from RAxML analysis of a concatenated ITS-LSU-SSU-*tef1-α* sequence dataset to represent the phylogenetic relationships of novel taxa in Parabambusicolaceae with other related families in Pleosporales. Bootstrap support values for ML equal to or greater than 70% and the Bayesian posterior probabilities equal to or higher than 0.95 PP are indicated above the nodes as ML/PP. Support values lower than 70% ML and 0.95 PP are indicated by a hyphen (-). Ex-type strains are in bold, and the new species is indicated in blue. The arrow in the figure is indicated the support value at the node.

**Figure 3 jof-08-00630-f003:**
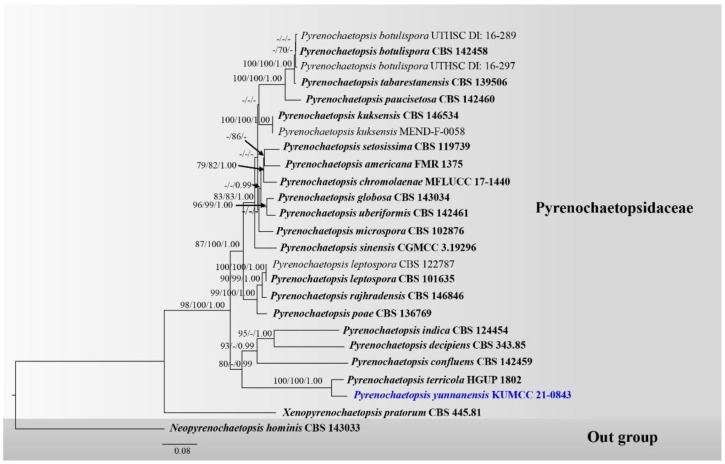
Phylogram generated from RAxML analysis of a concatenated LSU-ITS-*rpb2*-*tub2* sequence dataset to represent the phylogenetic relationships of a novel taxon in Pyrenochaetopsidaceae. Bootstrap support values for ML and MP equal to or greater than 70% and the Bayesian posterior probabilities equal to or higher than 0.95 PP are indicated above the nodes as ML/MP/PP. Support values lower than 70% ML/MP and 0.95 PP are indicated by a hyphen (-). Ex-type strains are in bold, and the new species is indicated in blue. The arrow is indicated the support value at the node.

**Figure 4 jof-08-00630-f004:**
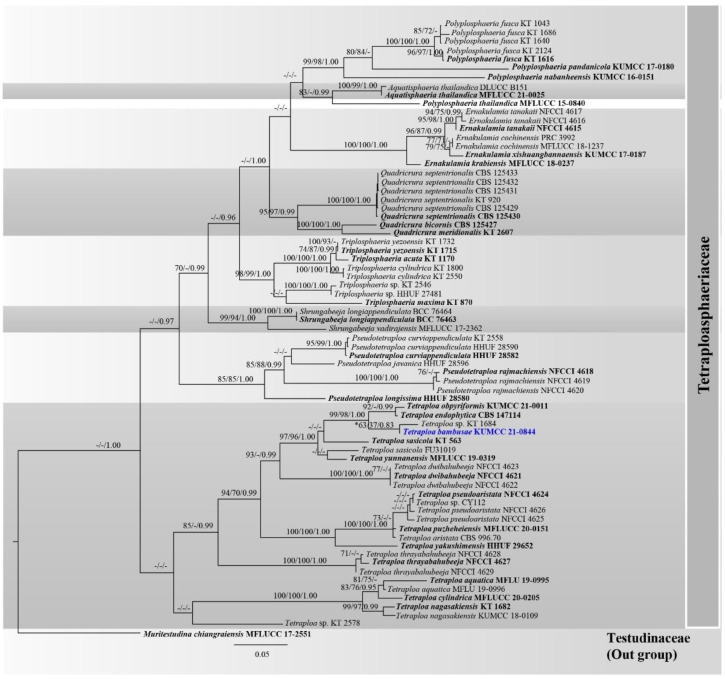
Phylogram generated from RAxML analysis of a concatenated LSU-ITS-SSU-*tub2*-*tef1-α* sequence dataset to represent the phylogenetic relationships of the novel taxon in Tetraploasphaeriaceae. Bootstrap support values for ML and MP equal to or greater than 70% and the Bayesian posterior probabilities equal to or higher than 0.95 PP are indicated above the nodes as ML/MP/PP. Support values lower than 70% ML/MP and 0.95 PP are indicated by a hyphen (-). Ex-type strains are in bold, and the new species is indicated in blue.

**Table 1 jof-08-00630-t001:** GenBank accession numbers used in the phylogenetic analyses. The ex-type cultures are indicated with superscript “^T^”, and the newly generated sequences are indicated in bold.

Taxon	Voucher/Strain No.	Family		GenBank Accession Number				
			ITS	LSU	SSU	*rpb2*	* tub2 *	* tef1-α *
*Anastomitrabeculia didymospora*	MFLUCC 16-0412 ^T^	Anast	NR_172008	MW412978	NG_073568	n/a	n/a	MW411338
*Anastomitrabeculia didymospora*	MFLUCC 16-0417	Anast	MW413897	MW413899	MW413898	n/a	n/a	MW411339
** *Anastomitrabeculia didymospora* **	**MFLUCC 11-0197**	**Anast**	**ON077079**	**ON077068**	**n/a**	**n/a**	**n/a**	**ON075062**
** *Anastomitrabeculia didymospora* **	**MFLUCC 11-0200**	**Anast**	**ON077080**	**ON077069**	**ON077074**	**ON075067**	**n/a**	**ON075063**
*Aquastroma magniostiolata*	KT 2485/HHUF 30122 ^T^	Parab	NR_153583	NG_056936	NG_061000	n/a	n/a	AB808486
*Aquatisphaeria thailandica*	MFLUCC 21-0025 ^T^	Tetra	MW890969	MW890763	MW890967	n/a	n/a	n/a
*Aquatisphaeria thailandica*	DLUCC B151	Tetra	n/a	MW890764	MW890968	n/a	n/a	n/a
*Bambusicola loculata*	MFLUCC 13-0856 ^T^	Bambu	NR_153609	NG_069267	NG_065061	KP761715	n/a	KP761724
*Bambusicola massarinia*	MFLUCC 11-0389 ^T^	Bambu	NR_121548	NG_058658	NG_061198	KP761716	n/a	KP761725
*Brevicollum hyalosporum*	MFLUCC 17-0071	Neohe	MG602204	MG602200	MG602202	n/a	n/a	MG739516
*Brevicollum hyalosporum*	MAFF 243400 ^T^	Neohe	NR_156334	NG_058715	NG_065123	LC271249	n/a	LC271245
*Brevicollum versicolor*	HHUF 30591 ^T^	Neohe	NR_156335	NG_058716	NG_065124	LC271250	n/a	LC271246
*Camarographium koreanum*	CBS 117159 ^T^	Macro	JQ044432	JQ044451	n/a	n/a	n/a	n/a
*Crassiparies quadrisporus*	KH111/HHUF 30409 ^T^	Neohe	NR_148185	NG_059028	NG_061267	n/a	n/a	n/a
*Crassiparies quadrisporus*	KT 2986/HHUF 30590	Neohe	LC271244	LC271241	LC271238	LC271252	n/a	LC271248
*Dendryphiella fasciculata*	MFLUCC 17-1074 ^T^	Dictyo	NR_154044	NG_059177	n/a	n/a	n/a	n/a
*Dictyocheirospora aquatica*	KUMCC 15-0305 ^T^	Dictyo	NR_154030	KY320513	n/a	n/a	n/a	n/a
*Dictyosporium elegans*	NBRC 32502	Dictyo	DQ018087	DQ018100	DQ018079	n/a	n/a	n/a
*Didymocrea sadasivanii*	CBS 438.65	Didymo	MH858658	DQ384103	DQ384074	n/a	n/a	n/a
*Didymosphaeria rubi-ulmifolii*	MFLUCC 14-0023 ^T^	Didymo	n/a	KJ436586	NG_063557	n/a	KJ939277	n/a
*Ernakulamia cochinensis*	PRC 3992	Tetra	LT964671	LT964670	n/a	n/a	LT964672	n/a
*Ernakulamia cochinensis*	MFLUCC 18-1237	Tetra	MT627670	MN913716	MT627670	n/a	n/a	n/a
*Ernakulamia krabiensis*	KUMCC 18-0237 ^T^	Tetra	NR_163341	NG_066314	NG_065780	MK434872	n/a	n/a
*Ernakulamia tanakaii*	NFCCI 4617	Tetra	MN937228	MN937210	n/a	n/a	MN938311	n/a
*Ernakulamia tanakaii*	NFCCI 4615 ^T^	Tetra	MN937229	MN937211	n/a	n/a	MN938312	n/a
*Ernakulamia tanakaii*	NFCCI 4616	Tetra	MN937227	MN937209	n/a	n/a	MN938310	n/a
*Ernakulamia xishuangbannaensis*	KUMCC 17-0187 ^T^	Tetra	MH275080	MH260314	MH260354	n/a	n/a	n/a
*Falciformispora lignatilis*	BCC 21117	Trema	KF432942	GU371826	GU371834	n/a	n/a	GU371819
*Halojulella avicenniae*	BCC 20173	Haloj	n/a	GU371822	GU371830	GU371786	n/a	GU371815
*Halojulella avicenniae*	BCC 18422	Haloj	n/a	GU371823	GU371831	GU371787	n/a	GU371816
*Halojulella avicenniae*	PUFD542	Haloj	MK028713	MK026757	MK026754	MN532682	n/a	n/a
*Helminthosporium aquaticum*	MFLUCC 15-0357 ^T^	Massa	NR_155170	NG_059656	NG_063601	n/a	n/a	n/a
*Katumotoa bambusicola*	KT 1517a/HHUF 28661 ^T^	Lenti	NR_154103	NG_059386	NG_060989	AB539095	n/a	AB539108
*Lonicericola fuyuanensis*	MFLU 19-2850 ^T^	Parab	NR_172419	NG_073809	NG_070329	n/a	n/a	MN938324
*Lonicericola hyaloseptispora*	KUMCC 18-0149 ^T^	Parab	NR_164294	NG_066434	NG_067680	n/a	n/a	n/a
*Lonicericola hyaloseptispora*	KUMCC 18-0150	Parab	MK098194	MK098200	MK098206	n/a	n/a	MK098210
*Macrodiplodiopsis desmazieri*	CBS 140062 ^T^	Macro	NR_132924	NG_058182	n/a	n/a	n/a	n/a
*Magnicamarosporium iriomotense*	KT 2822/HHUF 30125 ^T^	Sulca	NR_153445	NG_059389	NG_060999	n/a	n/a	AB808485
*Massarina eburnea*	CBS 473.64	Massa	OM337528	MH877786	GU296170	GU371732	n/a	GU349040
*Medicopsis romeroi*	CBS 252.60^T^	Neohe	NR_130697	NG_057800	NG_061069	KF015708	n/a	KF015678
*Medicopsis romeroi*	CBS 132878	Neohe	KF015658	KF015622	KF015648	KF015709	n/a	KF015682
*Medicopsis romeroi*	CBS 122784	Neohe	KF366447	EU754208	EU754109	KF015707	n/a	KF015679
*Melanomma pulvis-pyrius*	CBS 124080	Melan	MH863349	MH874873	GU456302	GU456350	n/a	GU456265
*Monodictys* sp.	JO 10	Parab	n/a	AB807552	AB797262	n/a	n/a	AB808528
*Monodictys* sp.	KH 331	Parab	n/a	AB807553	AB797263	n/a	n/a	AB808529
*Multilocularia bambusae*	MFLUCC 11-0180 ^T^	Parab	NR_148099	NG_059654	NG_061229	n/a	n/a	KU705656
*Multiseptospora thailandica*	MFLUCC 11-0183 ^T^	Parab	NR_148080	NG_059554	KP753955	n/a	n/a	KU705657
*Multiseptospora thailandica*	MFLUCC 11-0204	Parab	KU693447	KU693440	KU693444	n/a	n/a	KU705659
*Multiseptospora thailandica*	MFLUCC 12-0006	Parab	KU693448	KU693441	KU693445	n/a	n/a	KU705660
*Multiseptospora thysanolaenae*	MFLUCC 11-0202 ^T^	Parab	n/a	NG_059655	NG_063600	n/a	n/a	KU705658
*Muritestudina chiangraiensis*	MFLUCC 17-2551 ^T^	Testu	MG602247	MG602248	MG602249	MG602250	n/a	MG602251
*Neoaquastroma bauhiniae*	MFLUCC 16-0398 ^T^	Parab	NR_165217	NG_067814	NG_070696	MH028251	n/a	MH028247
*Neoaquastroma bauhiniae*	MFLUCC 17-2205	Parab	MH025953	MH023320	MH023316	MH028252	n/a	MH028248
*Neoaquastroma guttulatum*	MFLUCC 14-0917 ^T^	Parab	KX949739	KX949740	KX949741	n/a	n/a	KX949742
*Neoaquastroma krabiense*	MFLUCC 16-0419 ^T^	Parab	NR_165218	NG_067815	NG_067670	n/a	n/a	MH028249
*Neobambusicola strelitziae*	CBS 138869 ^T^	Sulca	NR_137945	NG_058125	n/a	n/a	n/a	MG976037
*Neohendersonia kickxii*	CBS 112403 ^T^	Neohe	NR_154248	NG_058264	n/a	n/a	n/a	n/a
*Neohendersonia kickxii*	CBS 122938	Neohe	KX820257	KX820268	n/a	n/a	n/a	n/a
*Neohendersonia kickxii*	CPC 24865	Neohe	KX820259	KX820270	n/a	n/a	n/a	n/a
*Neomultiseptospora yunnanensis*	KUMCC 21-0411 ^T^	Parab	OL898884	OL898925	OL898890	n/a	n/a	OL964282
*Neomultiseptospora yunnanensis*	KUN-HKAS 122240 ^T^	Parab	OL898885	OL898886	OL898891	n/a	n/a	OL964283
* Neopyrenochaetopsis hominis *	CBS 143033 ^T^	Pyren	LN880536	LN880537	n/a	LT593073	LN880539	n/a
*Palmiascoma gregariascomum*	MFLUCC 11-0175 ^T^	Bambu	NR_154316	NG_059557	KP753958	KP998466	n/a	n/a
*Parabambusicola aquatica*	MFLUCC 18-1140 ^T^	Parab	NR_171877	NG_073791	n/a	n/a	n/a	n/a
*Parabambusicola bambusina*	H 4321/MAFF 239462	Parab	LC014578	AB807536	AB797246	n/a	n/a	AB808511
*Parabambusicola bambusina*	KH 139/MAFF 243823	Parab	LC014579	AB807537	AB797247	n/a	n/a	AB808512
*Parabambusicola bambusina*	KT 2637/MAFF 243822	Parab	LC014580	AB807538	AB797248	n/a	n/a	AB808513
*Parabambusicola hongheensis*	KUMCC 21-0410 ^T^	Parab	OL898880	OL898921	OL898886	n/a	n/a	n/a
*Parabambusicola thysanolaenae*	KUMCC 18-0147 ^T^	Parab	NR_164044	NG_066435	NG_067681	n/a	n/a	MK098209
*Parabambusicola thysanolaenae*	KUMCC 18-0148	Parab	MK098193	MK098198	MK098202	n/a	n/a	MK098211
*Paraconiothyrium estuarinum*	CBS 109850 ^T^	Didym	NR_166007	MH874432	AY642522	LT854937	JX496355	n/a
*Paramonodictys hongheensis*	KUMCC 21-0343 ^T^	Parab	OL436229	OL436227	OL436232	n/a	n/a	OL505582
*Paramonodictys hongheensis*	KUMCC 21-0346	Parab	OL436235	OL436224	OL436225	n/a	n/a	OL505583
*Paramonodictys solitarius*	GZCC 20-0007 ^T^	Parab	MN901152	MN897835	MN901118	MT023015	n/a	MT023012
*Paramonodictys solitarius*	MFLUCC 17-2353	Parab	MT627707	MN913703	MT864299	n/a	n/a	MT954397
*Paramonodictys yunnanensis*	KUMCC 21-0337 ^T^	Parab	OL436231	OL436226	OL436230	n/a	n/a	OL505585
*Paramonodictys yunnanensis*	KUMCC 21-0347	Parab	OL436233	OL436228	OL436234	n/a	n/a	OL505586
** *Paramultiseptospora bambusae* **	**KUN-HKAS 122241A^T^**	**Parab**	**ON077075**	**ON077064**	**ON077070**	**n/a**	**n/a**	**ON075058**
** *Paramultiseptospora bambusae* **	**KUN-HKAS 122241B**	**Parab**	**ON077076**	**ON077065**	**ON077071**	**n/a**	**n/a**	**ON075059**
*Paraphaeosphaeria michotii*	MFLUCC 13-0349 ^T^	Didym	NR_155640	NG_059522	KJ939285	KP998465	n/a	n/a
*Paratrimmatostroma kunmingensis*	KUN-HKAS 102224A ^T^	Parab	MK098192	MK098196	MK098204	n/a	n/a	MK098208
*Paratrimmatostroma kunmingensis*	KUN-HKAS 102224B ^T^	Parab	MK098195	MK098201	MK098207	n/a	n/a	n/a
*Poaceascoma helicoides*	MFLUCC 11-0136 ^T^	Lenti	NR_154317	NG_059565	NG_061205	KP998460	n/a	KP998461
*Polyplosphaeria fusca*	KT 1043	Tetra	AB524788	AB524603	AB524462	n/a	AB524850	AB524819
*Polyplosphaeria fusca*	KT 1640	Tetra	AB524790	AB524605	AB524464	n/a	AB524852	AB524821
*Polyplosphaeria fusca*	KT 1616 ^T^	Tetra	AB524789	AB524604	AB524463	n/a	AB524851	AB524820
*Polyplosphaeria fusca*	KT 1686	Tetra	n/a	AB524606	AB524465	n/a	n/a	n/a
*Polyplosphaeria fusca*	KT 2124	Tetra	AB524791	AB524607	AB524466	n/a	AB524853	AB524822
*Polyplosphaeria nabanheensis*	KUMCC 16-0151 ^T^	Tetra	MH275078	MH260312	MH260352	n/a	MH412745	n/a
*Polyplosphaeria pandanicola*	KUMCC 17-0180 ^T^	Tetra	MH275079	MH260313	MH260353	n/a	n/a	n/a
*Polyplosphaeria thailandica*	MFLUCC 15-0840 ^T^	Tetra	KU248766	KU248767	n/a	n/a	n/a	n/a
*Pseudochaetosphaeronema larense*	CBS 640.73 ^T^	Macro	NR_132038	NG_057978	NG_061147	KF015706	n/a	KF015684
*Pseudocoleophoma calamagrostidis*	KT 3284/HHUF 30450 ^T^	Dicty	NR_154375	NG_059804	NG_061264	n/a	n/a	LC014614
*Pseudomonodictys tectonae*	MFLUCC 12-0552 ^T^	Parab	n/a	NG_059590	NG_061213	KT285572	n/a	KT285571
*Pseudotetraploa curviappendiculata*	HHUF 28582 ^T^	Tetra	AB524792	AB524608	AB524467	n/a	AB524854	AB524823
*Pseudotetraploa curviappendiculata*	KT 2558	Tetra	AB524794	AB524610	AB524469	n/a	AB524856	AB524825
*Pseudotetraploa curviappendiculata*	HHUF 28590	Tetra	AB524793	AB524609	AB524468	n/a	AB524855	AB524824
*Pseudotetraploa javanica*	HHUF 28596	Tetra	AB524795	AB524611	AB524470	n/a	AB524857	AB524826
*Pseudotetraploa longissima*	HHUF 28580 ^T^	Tetra	AB524796	AB524612	AB524471	n/a	AB524858	AB524827
*Pseudotetraploa rajmachiensis*	NFCCI 4619	Tetra	MN937222	MN937204	n/a	n/a	MN938305	n/a
*Pseudotetraploa rajmachiensis*	NFCCI 4618 ^T^	Tetra	MN937221	MN937203	n/a	n/a	MN938304	n/a
*Pseudotetraploa rajmachiensis*	NFCCI 4620	Tetra	MN937223	MN937205	n/a	n/a	MN938306	n/a
* Pyrenochaetopsis americana *	FMR 1375 ^T^	Pyren	LT592912	LN907368	n/a	LT593050	LT592981	n/a
** * Pyrenochaetopsis yunnanensis * **	**KUMCC 21-0843**	**Pyren**	**ON077077**	** ON077066 **	**ON077072**	**ON075066**	**ON075064**	**ON075060**
* Pyrenochaetopsis botulispora *	UTHSC:DI 16-289	Pyren	LT592941	LN907432	n/a	LT593080	LT593010	n/a
* Pyrenochaetopsis botulispora *	UTHSC:DI 16-297	Pyren	LT592945	LN907440	n/a	LT593084	LT593014	n/a
* Pyrenochaetopsis botulispora *	CBS 142458 ^T^	Pyren	LT592946	LN907441	n/a	LT593085	LT593015	n/a
*Pyrenochaetopsis chromolaenae*	MFLUCC: 17-1440 ^T^	Pyren	MT214378	MT214472	NG_070172	MT235827	n/a	MT235790
* Pyrenochaetopsis confluens *	CBS 142459 ^T^	Pyren	LT592950	LN907446	n/a	LT593089	LT593019	n/a
* Pyrenochaetopsis decipiens *	CBS 343.85 ^T^	Pyren	LT623223	GQ387624	NG_065569	LT623280	LT623240	n/a
* Pyrenochaetopsis globosa *	CBS 143034 ^T^	Pyren	LT592934	LN907418	n/a	LT593072	LT593003	n/a
* Pyrenochaetopsis indica *	CBS 124454 ^T^	Pyren	LT623224	GQ387626	GQ387565	LT623281	LT623241	n/a
* Pyrenochaetopsis kuksensis *	CBS 146534/MeND-F-57 ^T^	Pyren	MT371092	MT371397	n/a	MT372656	MT372662	n/a
* Pyrenochaetopsis kuksensis *	MeND-F-58	Pyren	MT371093	MT371398	n/a	MT372657	MT372663	n/a
* Pyrenochaetopsis leptospora *	CBS 101635 ^T^	Pyren	JF740262	GQ387627	NG_063097	LT623282	LT623242	MF795881
* Pyrenochaetopsis leptospora *	CBS 122787	Pyren	LT623225	EU754151	n/a	LT623283	LT623243	n/a
* Pyrenochaetopsis microspora *	CBS 102876 ^T^	Pyren	LT592899	LN907341	NG_065571	LT593037	LT592968	n/a
* Pyrenochaetopsis paucisetosa *	CBS 142460 ^T^	Pyren	LT592897	LN907336	n/a	LT593035	LT592966	n/a
* Pyrenochaetopsis poae *	CBS 136769 ^T^	Pyren	KJ869117	KJ869175	n/a	LT623286	KJ869243	n/a
* Pyrenochaetopsis rajhradensis *	CBS 146846 ^T^	Pyren	MT853115	MT853182	n/a	MT857727	MT857726	MT857725
* Pyrenochaetopsis setosissima *	CBS 119739 ^T^	Pyren	LT623227	GQ387632	n/a	LT623285	LT623245	n/a
* Pyrenochaetopsis sinensis *	CGMCC 3.19296 ^T^	Pyren	MK348586	MK348581	n/a	MK355077	MK348221	n/a
* Pyrenochaetopsis tabarestanensis *	CBS 139506/IBRC: M 30051 ^T^	Pyren	KF730241	KF803343	NG_065034	n/a	KX789523	n/a
* Pyrenochaetopsis terricola *	HGUP 1802 ^T^	Pyren	MH697394	MH697393	n/a	MH697395	MH697392	n/a
* Pyrenochaetopsis uberiformis *	CBS 142461/FMR 13769 ^T^	Pyren	LT592935	LN907420	n/a	LT593074	LT593004	n/a
*Quadricrura bicornis*	CBS 125427 ^T^	Tetra	AB524797	AB524613	AB524472	n/a	AB524859	AB524828
*Quadricrura meridionalis*	KT 2607 ^T^	Tetra	AB524798	AB524614	AB524473	n/a	AB524860	AB524829
*Quadricrura septentrionalis*	KT 920	Tetra	AB524801	AB524617	AB524476	n/a	AB524863	AB524832
*Quadricrura septentrionalis*	CBS 125429	Tetra	AB524799	AB524615	AB524474	n/a	AB524861	AB524830
*Quadricrura septentrionalis*	CBS 125431	Tetra	AB524802	AB524618	AB524477	n/a	AB524864	AB524833
*Quadricrura septentrionalis*	CBS 125432	Tetra	AB524803	AB524619	AB524478	n/a	AB524865	AB524834
*Quadricrura septentrionalis*	CBS 125433	Tetra	AB524804	AB524620	AB524479	n/a	AB524866	AB524835
*Quadricrura septentrionalis*	CBS 125430 ^T^	Tetra	AB524800	AB524616	AB524475	n/a	AB524862	AB524831
*Scolecohyalosporium submersum*	KUMCC 21-0412 ^T^	Parab	OL898883	OL898924	OL898889	n/a	n/a	OL964281
*Scolecohyalosporium submersum*	KUMCC 21-0413	Parab	OL898881	OL898922	OL898887	n/a	n/a	OL964279
*Scolecohyalosporium submersum*	KUN-HKAS 122242	Parab	OL898882	OL898923	OL898888	n/a	n/a	OL964280
*Setoseptoria phragmitis*	CBS 114802 ^T^	Lenti	KF251249	KF251752	n/a	KF252254	KF252732	KF253199
*Shrungabeeja longiappendiculata*	BCC 76463 ^T^	Tetra	KT376474	KT376472	KT376471	n/a	n/a	n/a
*Shrungabeeja longiappendiculata*	BCC 76464	Tetra	KT376475	KT376473	n/a	n/a	n/a	n/a
*Shrungabeeja vadirajensis*	MFLUCC 17-2362	Tetra	MT627681	MN913685	n/a	n/a	n/a	n/a
*Spegazzinia tessarthra*	SH 287	Didym	n/a	AB807584	AB797294	n/a	n/a	AB808560
*Sulcatispora acerina*	KT 2982 ^T^	Sulca	LC014597	LC014610	LC014605	n/a	n/a	LC014615
*Sulcatispora berchemiae*	KT 1607/ HHUF 29097 ^T^	Sulca	NR_153444	NG_059390	NG_064843	n/a	n/a	AB808509
*Tetraploa aquatica*	MFLU 19-0995 ^T^	Tetra	MT530448	MT530452	n/a	n/a	n/a	n/a
*Tetraploa aquatica*	MFLU 19-0996	Tetra	MT530449	MT530453	MT530454	n/a	n/a	n/a
*Tetraploa aristata*	CBS 996.70	Tetra	AB524805	AB524627	AB524486	n/a	AB524867	AB524836
** *Tetraploa bambusae* **	**KUMCC 21-0844**	**Tetra**	**ON077078**	**ON077067**	**ON077073**	**n/a**	**ON075065**	**ON075061**
*Tetraploa* *cylindrica*	KUMCC 20-0205 ^T^	Tetra	MT893205	MT893204	MT893203	n/a	MT899418	MT899417
*Tetraploa dashaoensis*	KUMCC 21-0010 ^T^	Tetra	OL473549	OL473555	OL473556	n/a	OL505601	OL505599
*Tetraploa dwibahubeeja*	NFCCI 4621 ^T^	Tetra	MN937225	MN937207	n/a	n/a	MN938308	n/a
*Tetraploa dwibahubeeja*	NFCCI 4622	Tetra	MN937224	MN937206	n/a	n/a	MN938307	n/a
*Tetraploa dwibahubeeja*	NFCCI 4623	Tetra	MN937226	MN937208	n/a	n/a	MN938309	n/a
*Tetraploa endophytica*	CBS 147114 ^T^	Tetra	n/a	MW659165	KT270279	n/a	n/a	MW659821
*Tetraploa nagasakiensis*	KUMCC 18-0109	Tetra	MK079890	MK079891	MK079888	n/a	n/a	n/a
*Tetraploa nagasakiensis*	KT 1682 ^T^	Tetra	AB524806	AB524630	AB524489	n/a	AB524868	AB524837
*Tetraploa obpyriformis*	KUMCC 21-0011 ^T^	Tetra	OL473558	OL473554	OL473557	n/a	OL505600	OL505598
*Tetraploa pseudoaristata*	NFCCI 4624 ^T^	Tetra	MN937232	MN937214	n/a	n/a	MN938315	n/a
*Tetraploa pseudoaristata*	NFCCI 4625	Tetra	MN937230	MN937212	n/a	n/a	MN938313	n/a
*Tetraploa pseudoaristata*	NFCCI 4626	Tetra	MN937231	MN937213	n/a	n/a	MN938314	n/a
*Tetraploa puzheheiensis*	MFLUCC 20-0151 ^T^	Tetra	MT627744	MT627655	n/a	n/a	n/a	n/a
*Tetraploa sasicola*	FU31019	Tetra	MN937236	MN937218	n/a	n/a	n/a	n/a
*Tetraploa sasicola*	KT 563 ^T^	Tetra	AB524807	AB524631	AB524490	n/a	AB524869	AB524838
*Tetraploa* sp.	KT 1684	Tetra	n/a	AB524628	AB524487	n/a	n/a	n/a
*Tetraploa* sp.	KT 2578	Tetra	n/a	AB524629	AB524488	n/a	n/a	n/a
*Tetraploa* sp.	CY112	Tetra	HQ607964	n/a	n/a	n/a	n/a	n/a
*Tetraploa thrayabahubeeja*	NFCCI 4627 ^T^	Tetra	MN937235	MN937217	n/a	n/a	MN938318	n/a
*Tetraploa thrayabahubeeja*	NFCCI 4628	Tetra	MN937233	MN937215	n/a	n/a	MN938316	n/a
*Tetraploa thrayabahubeeja*	NFCCI 4629	Tetra	MN937234	MN937216	n/a	n/a	MN938317	n/a
*Tetraploa yakushimensis*	KT 1906/HHUF 29652 ^T^	Tetra	NR_119405	NG_042330	NG_064836	n/a	AB524870	AB524839
*Tetraploa yunnanensis*	MFLUCC 19-0319 ^T^	Tetra	MT627743	MN913735	MT864341	MT878451	n/a	n/a
*Trematosphaeria grisea*	CBS 332.50 ^T^	Trema	NR_132039	NG_057979	NG_062930	KF015720	n/a	KF015698
*Trematosphaeria_pertusa*	CBS 122368 ^T^	Trema	NR_132040	NG_057809	n/a	n/a	n/a	n/a
*Tingoldiago graminicola*	KH 68 ^T^	Lenti	LC014598	AB521743	AB521726	n/a	n/a	AB808561
*Triplosphaeria acuta*	KT 1170 ^T^	Tetra	AB524809	AB524633	AB524492	n/a	AB524871	AB524840
*Triplosphaeria cylindrica*	KT 1800	Tetra	AB524810	AB524635	AB524494	n/a	AB524872	AB524841
*Triplosphaeria cylindrica*	KT 2550	Tetra	AB524811	AB524636	AB524495	n/a	AB524873	AB524842
*Triplosphaeria maxima*	KT 870/HHUF 29390 ^T^	Tetra	AB524812	AB524637	AB524496	n/a	AB524874	AB524843
*Triplosphaeria* sp.	HHUF 27481	Tetra	AB524815	AB524640	AB524499	n/a	AB524877	AB524846
*Triplosphaeria* sp.	KT 2546	Tetra	AB524816	AB524641	AB524500	n/a	AB524878	AB524847
*Triplosphaeria yezoensis*	KT 1715 ^T^	Tetra	AB524813	AB524638	AB524497	n/a	AB524875	AB524844
*Triplosphaeria yezoensis*	KT 1732	Tetra	AB524814	AB524639	AB524498	n/a	AB524876	AB524845
*Trematosphaeria grisea*	CBS 332.50 ^T^	Trema	NR_132039	NG_057979	NG_062930	KF015720	n/a	KF015698
*Trematosphaeria pertusa*	CBS 122368 ^T^	Trema	NR_132040	NG_057809	n/a	n/a	n/a	n/a
* Xenopyrenochaetopsis pratorum *	CBS 445.81/FMR 14878 ^T^	Pyren	JF740263	GU238136	NG_062792	KT389671	KT389846	n/a

Abbreviations: BCC: BIOTEC Culture Collection, Bangkok, Thailand; BCRC: FU: Bioresource Collection and Research Center Collection, Taiwan; CBS: the Westerdijk Fungal Biodiversity Institute, Utrecht, The Netherlands; CGMCC: China General Microbiological Culture Collection Center; CPC: Collection of Pedro Crous housed at CBS; DLUCC: Dali University Culture Collection, Yunnan, China; FMR: Facultat de Medicina, Universitat Rovira i Virgili, Reus, Spain; GZCC: Guizhou Culture Collection, Guizhou, China; H: University of Helsinki, Helsinki, Finland; HGUP: Herbarium of Department of Plant Pathology, Guizhou University, Guizhou, China; HHUF: the Herbarium of Hirosaki University Fungi, Aomori, Japan; IBRC: M: Herbarium of the Plant bank, Iranian Biological Resource Center; KH: K. Hirayama; KT: Kazuaki Tanaka, Japan; KUMCC: Kunming Institute of Botany Culture Collection, Yunnan, China; KUN-HKAS: Herbarium of Cryptogams Kunming Institute of Botany Academia Sinica, Yunnan, China; MeND-F: Fungal Collection of Mendeleum—Institute of Genetics, Mendel University, Czech Republic; MAFF: the National Institute of Agrobiological Sciences, Japan; MFLU: the Herbarium of Mae Fah Luang University Chiang Rai, Thailand; MFLUCC: Mae Fah Luang University Culture Collection, Chiang Rai, Thailand; NBRC: Biological Resource Center, National Institute of Technology and Evaluation, Chiba, Japan; NFCCI: National Fungal Culture Collection of India, Maharashtra, India; PRC: the Herbarium of the Charles University, Prague, Czech Republic; PUFU: Culture Collection at Pondicherry University, Puducherry, India; SH: S. Hughes; UTHSC: Fungus Testing Laboratory at the University of Texas Health Science Center, San Antonio, Texas, USA. Abbreviations of families: Anast: Anastomitrabeculiaceae; Bambu: Bambusicolaceae; Dicty: Dictyosporiaceae; Didym: Didymosphaeriaceae; Haloj: Halojulellaceae; Lenti: Lentitheciaceae; Macro: Macrodiplodiopsidaceae; Massa: Massarinaceae; Melan: Melanommataceae; Neohe: Neohendersoniaceae; Parab: Parabambusicolaceae; Pyren: Pyrenochaetopsidaceae; Sulca: Sulcatisporaceae; Testu: Testudinaceae; Tetra: Tetraploasphaeriaceae; Trema: Trematosphaeriaceae.

**Table 2 jof-08-00630-t002:** The best nucleotide substitution model for each locus based on the Akaike Information Criterion (AIC) generated by MrModeltest v. 2.3.

Phylogenetic Analyses	Nucleotide Substitution Models					
	ITS	LSU	SSU	*rpb2*	* tef1-α *	*tub2*
A1: Anastomitrabeculiaceae	SYM+G	GTR+I+G	GTR+I	n/a	GTR+I+G	n/a
A2: Parabambusicolaceae	GTR+I+G	GTR+I+G	HKY+I+G	n/a	GTR+I+G	n/a
A3: Pyrenochaetopsidaceae	SYM+G	GTR+I	n/a	SYM+I+G	GTR+G	n/a
A4: Tetraploasphaeriaceae	GTR+I+G	GTR+I+G	GTR+I+G	n/a	HKY+I+G	GTR+I+G

## Data Availability

All data availability was mentioned in the manuscript. The novel taxa were registered in Index Fungorum (http://www.indexfungorum.org/Names/Names.asp, accessed on 13 May 2022) including Index Fungorum numbers IF 554966, IF 554968, IF 554979 and IF 554987. Final alignment and phylogenetic tree were deposited in TreeBase (https://www.treebase.org/, accessed on 25 March 2022) with submission ID: 29589, 29590, 29592 and 29593) and the newly generated sequences were deposited in GenBank (https://www.ncbi.nlm.nih.gov/genbank/submit/, accessed on 28 March 2022) followed as ITS: ON077079, ON077080, ON077075, ON077076, ON077077, ON077078; LSU: ON077068, ON077069, ON077064, ON077065, ON077066, ON077067; SSU: ON077074, ON077070, ON077071, ON077072, ON077073; *rpb2*: ON075067, ON075066; *tef1*-α: ON075062, ON075063, ON075058, ON075059, ON075060, ON075061; *tub2*: ON075064, ON075065.

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
