# Peer review of "Bambusicolous Fungi in Pleosporales: Introducing Four Novel Taxa and a New Habitat Record for Anastomitrabeculia didymospora"

_jof, 2022, doi:10.3390/jof8060630_

Round 1

Reviewer 1 Report

The authors are advised to improve the manuscript in terms of adequate language levels as well as research paper structure.

I recommend expanding the introduction section with more updated literature

English should be improved; grammar need enhancement in many sentences and paragraphs.

All figures are needed resolution enhancement.

Figures is not in printable quality. Also, some portions of the texts are losing their readability while sizing the image as per the text area. Kindly provide a better quality figure and increase the size of the figures.

Please check the References in-text and end-list for uniformity in style.

The conclusion you have provided is quite brief and provides sufficient feedback on the main objectives of your study.

Author Response

Dear reviewer,

Thank you very much for your invaluable comment and feedback.

Please see our response in the attachment.

Best regards

Reviewer 2 Report

The article provides very interesting information. Authors on the basis of multigene phylogenetic analyses present novel taxa: three new species (Paramultiseptospora bambusae sp. nov., Pyrenochaetopsis yunnanensis sp. nov. and Tetraploa bambusae sp. nov.) and new genus Paramultiseptospora gen. nov. of Pleosporales. In addition, authors present a new habitat record for Anastomitrabeculia didymospora, which was found on bamboo twigs in terrestrial habitats for the first time. Detailed morphological descriptions of novel taxa including phylogenetic trees of each family and very good illustrations are provided. I think that results of this study are new and important for taxonomy of Pleosporales, good enough analysed and I recommend publishing this paper in ‘Journal of Fungi’ after minor changes and corrections. Comments and questions are listed below.

Comments to the Author:

Abstract

Line 33. Please clarify on which genes was based multigene phylogenetic analyses.

Line 34. Presented in this manuscript novel taxa don’t belong to family Anastomitrabeculiaceae. To this family belong Anastomitrabeculia didymospora Bhunjun, Phukhams. & K.D. Hyde , which is not a new species, it was described in 2021. Please correct this part.

Keywords: I propose to reject names of novel taxa and include : “multigene; phylogeny, morphology’

Introduction

Line 71. ‘……Pleosporales – Latin name should be in italic Font.

Line 79-87. Please correct this part. It is not necessary to separate families of Pleosporales to which belong bambusicolous fungi. This information you can present in one sentence.

            Materials and methods

Line 93. Please clarify the years when samples were collected.

Line 188. Table 1. You note that “the newly generated sequences are indicated in bold.” But it is not clear for Anastomitrabeculia didymospora , accession numbers in the table are very close to each other, and it is difficult to separate them.

Results

Phylogeny

Line 236, 238. ……Anastomitrabeculia didymospora - must be in italic. Please check all Latin names of species, they must be in italic.

Line 251. Melanomma pulvis-pyrius  - italic

Line 276, 298. Pyrenochaetopsis yunnanensis, P. terricola – italic

Line 300. P. confluens, P. decipiens, P. indica – italic

Line 309. Tetraploa bambusae - italic

Line 33—335. Latin names must be in italic

Taxonomy

Line 387-388. What about sporulation on PDA?

Line 425. Correct the sentence ‚Etymology: Referring to the genus has a close phylogenetic relationship with Multiseptospora  to „ ...Reffering to relation with phylogenetically close genus Multiseptospora

Line 474. Latin name of new species must be in bold and italic - Paramultiseptospora bambusae

Line 476 Correct „.........., of on which the species was collected.‘

Line 494. Did this fungus was not isolated on PDA?

Line 501. Please note that DNA was extracted from fruitbodies.

Line 505-506. ), „......and is identical to “Pleosporales sp. strain 1192”(95.54% similarity, Identities = 407/426, with no gap). I think that it is not ‚ identicalwhen the similarity is only 95.54%

Line 662. „Conidiophores up to 40–130 μm long, (1.5–)2–3.5 μm wide, micronematous,..‘ If conidiophores are up to 40-130 μm theyare not micronematous.

Line 676. What about sporulation on PDA?

Line 731-732. „...Pyrenochaetopsis yunnanensis sp.nov., and Tetraploa bambusae sp. nov.,“

Line 757. (h) Culture characteristics on PDA after one week.

Line 781. It is better use „...rarely detected"

References

Line 1158, 1164, 1176.  Please check all names of journals, they must be in italic

Author Response

(The authors gave the same response as above.)
